# Desmin Knock-Out Cardiomyopathy: A Heart on the Verge of Metabolic Crisis

**DOI:** 10.3390/ijms231912020

**Published:** 2022-10-10

**Authors:** Barbara Elsnicova, Daniela Hornikova, Veronika Tibenska, David Kolar, Tereza Tlapakova, Benjamin Schmid, Markus Mallek, Britta Eggers, Ursula Schlötzer-Schrehardt, Viktoriya Peeva, Carolin Berwanger, Bettina Eberhard, Hacer Durmuş, Dorothea Schultheis, Christian Holtzhausen, Karin Schork, Katrin Marcus, Jens Jordan, Thomas Lücke, Peter F. M. van der Ven, Rolf Schröder, Christoph S. Clemen, Jitka M. Zurmanova

**Affiliations:** 1Department of Physiology, Faculty of Science, Charles University, 128 00 Prague, Czech Republic; 2Department of Cell Biology, Faculty of Science, Charles University, 128 00 Prague, Czech Republic; 3Optical Imaging Center Erlangen, Friedrich-Alexander University Erlangen-Nürnberg, 91058 Erlangen, Germany; 4MVZ Dr. Eberhard & Partner Dortmund, 44137 Dortmund, Germany; 5Medizinisches Proteom-Center, Medical Faculty, Ruhr-University Bochum, 44801 Bochum, Germany; 6Medical Proteome Analysis, Center for Proteindiagnostics (PRODI), Ruhr-University Bochum, 44801 Bochum, Germany; 7Department of Ophthalmology, University Hospital Erlangen, Friedrich-Alexander University Erlangen-Nürnberg, 91054 Erlangen, Germany; 8Division of Neurochemistry, Institute of Experimental Epileptology and Cognition Research, University of Bonn, 53127 Bonn, Germany; 9Institute of Aerospace Medicine, German Aerospace Center (DLR), Linder Höhe, 51147 Cologne, Germany; 10Department of Neurology, Istanbul Faculty of Medicine, Istanbul University, 34093 Istanbul, Turkey; 11Institute of Neuropathology, University Hospital Erlangen, Friedrich-Alexander University Erlangen-Nürnberg, 91054 Erlangen, Germany; 12Chair of Aerospace Medicine, Medical Faculty, University of Cologne, 50931 Cologne, Germany; 13University Hospital of Pediatrics and Adolescent Medicine, St. Josef-Hospital, Ruhr-University Bochum, 44791 Bochum, Germany; 14Department of Molecular Cell Biology, Institute for Cell Biology, University of Bonn, 53121 Bonn, Germany; 15Institute of Vegetative Physiology, Medical Faculty, University of Cologne, 50931 Cologne, Germany

**Keywords:** desmin, desminopathy, cardiomyopathy, mitochondriopathy, desmin knock-out metabolism, glucose, fatty acid, amino acid, creatine kinase, mitochondria

## Abstract

Desmin mutations cause familial and sporadic cardiomyopathies. In addition to perturbing the contractile apparatus, both desmin deficiency and mutated desmin negatively impact mitochondria. Impaired myocardial metabolism secondary to mitochondrial defects could conceivably exacerbate cardiac contractile dysfunction. We performed metabolic myocardial phenotyping in left ventricular cardiac muscle tissue in desmin knock-out mice. Our analyses revealed decreased mitochondrial number, ultrastructural mitochondrial defects, and impaired mitochondria-related metabolic pathways including fatty acid transport, activation, and catabolism. Glucose transporter 1 and hexokinase-1 expression and hexokinase activity were increased. While mitochondrial creatine kinase expression was reduced, fetal creatine kinase expression was increased. Proteomic analysis revealed reduced expression of proteins involved in electron transport mainly of complexes I and II, oxidative phosphorylation, citrate cycle, beta-oxidation including auxiliary pathways, amino acid catabolism, and redox reactions and oxidative stress. Thus, desmin deficiency elicits a secondary cardiac mitochondriopathy with severely impaired oxidative phosphorylation and fatty and amino acid metabolism. Increased glucose utilization and fetal creatine kinase upregulation likely portray attempts to maintain myocardial energy supply. It may be prudent to avoid medications worsening mitochondrial function and other metabolic stressors. Therapeutic interventions for mitochondriopathies might also improve the metabolic condition in desmin deficient hearts.

## 1. Introduction

In heart failure, cardiac contractile dysfunction and impaired mitochondrial function are intimately related [1]. Damage to the contractile apparatus promotes secondary changes in mitochondrial function and oxidative metabolism, thus, aggravating cardiac disease. Conversely, mitochondrial disease may produce secondary contractile dysfunction and heart failure [2]. Dysfunction in proteins affecting both myocardial contraction and mitochondrial function could conceivably produce a ‘double whammy’ on the heart. One such disease in which a primary cellular defect is related to a cytoskeletal protein with secondary changes in mitochondrial structure and number is desminopathy [3,4,5]. Desminopathies comprise a group of rare cardiomyopathies and myopathies caused by mutations of the human desmin gene (*DES*) on chromosome 2q35 [6,7]. More than 120 disease-causing *DES* mutations have been described ([8,9,10], Human Intermediate Filament Database, http://www.interfil.org (accessed on 5 October 2022)), which give rise to autosomal-dominant, autosomal-recessive, and sporadic desminopathies. Autosomal-dominantly inherited desminopathies—by far the most frequently encountered genetic form—usually manifest with signs of cardiac or skeletal muscle pathology [9,11,12]. In the very rare recessive desminopathies, a subset of *DES* mutations leads to desmin deficiency [13,14,15,16].

The deleterious effects of mutant desmin or desmin deletion in human striated muscles is closely linked to desmin’s multiple functions. Desmin is the principal intermediate filament component of the extrasarcomeric cytoskeleton in striated muscle cells, which is already expressed in the early stages of myogenesis [17,18]. Human desmin, a 470 amino acid protein with a molecular weight of 53.5 kDa, interlinks individual myofibrils at the level of Z-discs. Furthermore, desmin attaches the whole myofibrillar apparatus to sarcolemmal adhesion sites, myonuclei, intercalated disks, myotendinous and neuromuscular junctions, and tethers the extrasarcomeric cytoskeleton to mitochondria [3,18,19,20,21,22]. In desminopathies, faulty desmin expression inflicts a multitude of aberrations, thereby negatively affecting the mechanical stability of muscle cells, myofibrillar spatial organization with subsequently impaired force generation, and the structure and function of intercalated discs, neuromuscular junctions, and the mitochondrial network [4,8,11,12,14,23,24,25,26,27,28,29,30,31,32,33,34,35].

Previous studies repeatedly showed that desmin deletion is associated with mitochondrial dysfunction in striated muscle cells [3,4,5,36]. The finding led to the concept that a lack of desmin also perturbs metabolism which in turn further worsens muscle function. To address this issue in more detail, we performed comprehensive morphological, biochemical, and proteomic analyses of left ventricular cardiac tissue from desmin knock-out mice. Notably, thirteen years before the first description of patients lacking desmin [13], two independent research groups had reported their first analyses of desmin knock-out mice [20,21], which closely mirror the human pathology of autosomal-recessive desminopathies with a lack of desmin protein [13,14,15,16]. Our present work now reveals that beyond structural and functional changes in mitochondria, desmin deletion induces profound myocardial metabolic dysfunction including altered glucose, fatty acid, and amino acid metabolism.

## 2. Results

Previous work demonstrated that desmin deletion induces multiple structural alterations affecting the myofibrillar cytoarchitecture [20,21,37], the neuromuscular endplate [14,27,35], and mitochondria [3,4,5,31]. The common denominator of this diverse pathology is a defective extrasarcomeric cytoskeleton, which lacks its main component desmin. Here, we aimed to delineate the multi-level metabolic effects of the lack of desmin in left ventricular cardiac tissue (Figure 1a) derived from six-month-old desmin knock-out mice and their wild-type siblings by primarily studying mitochondrial bulk, distribution, ultrastructure, and enzyme activities and protein levels related to fatty acid, glucose and amino acid metabolism, and creatine kinase isoforms.

### 2.1. Desmin Deficiency Is Associated with Cardiac Mitochondrial Defects and Altered Fatty and Amino Acid Metabolism

In a first step, we determined three-dimensional mitochondrial distribution using image processing of confocal Z-stacks of left ventricular cardiac tissue sections stained for the outer mitochondrial membrane component voltage-dependent anion channel (VDAC1) (Figure 1b). Compared to cardiomyocytes from wild-type littermates, desmin knock-out cells displayed a rarefied mitochondrial network with significant numerical reduction in VDAC1-positive mitochondrial contact sites (Figure 1b,d). These findings suggest a reduction in the total number of mitochondria, which is further supported by the significantly reduced mitochondrial DNA (mtDNA) copy number (Figure 1e) without evidence of mtDNA deletions (Figure 1c). Furthermore, there was a numerical but statistically non-significant reduction of malate dehydrogenase (Figure 1f) and citrate synthase (Figure 1g) enzyme activities in desmin knock-out mice. Next, we aimed to enrich mitochondrial fractions for further biochemical analysis. However, repeated attempts failed due to the apparently very high fragility, i.e., loss of respiration ability and swelling, the latter observed via decreasing absorbance at 520 nm of the extracted mitochondria from the desmin knock-out cardiac tissue. Further evidence of mitochondrial pathology was provided from our ultrastructural analyses that revealed coarsened mitochondrial cristae in cardiac tissue of desmin knock-out mice (Figure 1h). This ultrastructural abnormality was only seen in electron microscopy images derived from desmin knock-out mice, where it was present in the majority of recordings. In addition to morphological alterations of mitochondria, electron microscopic analysis also showed markedly increased intermyofibrillar, electron-dense lipofuscin deposits in the desmin knock-out cardiac tissue (Figure 2a). However, Oil red O and PAS stains of additional left ventricular cardiac tissue sections did not differ between knock-out and wild-type sections. Thus, on the light microscopy level, there was neither obvious accumulation of lipofuscin nor of glycogen. Notably, immunoblotting revealed significantly decreased fatty acid transporter CD36 abundance (Figure 2b), and oxygen consumption measurements demonstrated significantly reduced octanoyl-carnitine-stimulated beta-oxidation (Figure 2c) in desmin knock-out cardiac tissue homogenates. Alterations in mitochondrial fatty acid metabolism were also mirrored by mass spectrometry-based acylcarnitine quantitation in dried whole blood samples. In desmin knock-out mice, this analysis showed significantly increased concentrations of multiple acylcarnitines comprising butyryl- and hydroxy-butyryl-carnitine (C4, C4OH), isovaleryl-carnitine (C5), octanoyl-carnitine (C8), tetradecanoyl-carnitine (C14), hydroxy-hexadecanoyl-carnitine (C16OH), and octanoyl-carnitine (C18) (Figure 2e) in conjunction with a significantly increased C8/C10-carnitine ratio (Table 1). Levels of propionyl-carnitine (C3), palmitoyl-carnitine (C16), 3OH-hexadecenoyl-carnitine (C16:1OH), and oleoyl-carnitine (C18:1) were numerically increased, however, they failed to reach statistical significance (Table 1). Notably, clinical chemistry data derived from a previously published patient (patient 2, [14]), a case of the very rare desminopathy subform with a lack of desmin, showed blood acylcarnitine levels in a normal range, but a moderately increased C8/C10-carnitine ratio (0.980; normal range, 0.000 to 0.600). The mass spectrometry analysis of the murine dried whole blood samples further revealed markedly elevated levels of the branched-chain amino acids valine, isoleucine, and leucine (Figure 2d; Table 1) as well as of two aromatic amino acids, phenylalanine and tryptophan (Table 1); the latter three, however, failed to reach statistical significance (Table 1).

### 2.2. Concomitant Changes in Myocardial Glucose Metabolism

To gain insight into the glucose metabolism of desmin knock-out left ventricular cardiac tissue, we first performed immunoblots addressing the expression of glucose transporter type 1 (GLUT1, Figure 3a) and type 4 (GLUT4, (Figure 3d). While the protein amount of GLUT1 was significantly higher in desmin knock-out mice, the level of GLUT4 remained unchanged. It is, however, noteworthy that the apparent molecular weight of GLUT4 was slightly increased in the desmin knock-out genotype (Figure 3d). The quantitative analysis of the subcellular localization of GLUT1 (Figure 3b,c) and GLUT4 (Figure 3e,f) in relation to the WGA-stained sarcolemma revealed significant differences between desmin knock-out mice and their wild-type siblings. In addition, qualitative visual examination of the signals of both glucose transporters depicted a localization of both GLUT1 and GLUT4 underneath the sarcolemma in desmin knock-out cardiomyocytes in contrast to their colocalization with the sarcolemma in wild-type tissue (Figure 3g). Subsequently, we investigated hexokinase and phosphofructokinase, the rate-limiting enzymes of glycolysis. The activity of hexokinase was significantly elevated in the desmin knock-out condition (Figure 4a). Immunoblot analysis further showed a significantly increased amount of hexokinase isoform 1 but not isoform 2 in the cardiac tissue homogenates of desmin knock-out mice (Figure 4b,c). In light of the notion that increased association of hexokinase isoform 1 with the outer mitochondrial membrane facilitates oxidative glucose utilization and that hexokinase isoform 2 additionally exerts a strong antiapoptotic activity via stabilization of the mitochondrial permeability transition pore [38], hexokinase-1, but not hexokinase-2, further displayed a significantly increased degree of colocalization with the mitochondrial compartment in desmin knock-out cardiomyocytes (Figure 4d,e).

### 2.3. Imbalance in the Creatine Kinase System

The creatine kinase system comprising one mitochondrial and two cytosolic creatine kinase isoforms in cardiomyocytes [39] enables the transfer of high-energy phosphate compounds between ATP production compartments, i.e., oxidative phosphorylation and glycolysis, and ATPases, thus serving as an energy buffer. Immunoblotting revealed an imbalance between creatine kinase isoforms in desmin knock-out cardiomyocytes with a significant reduction in mitochondrial creatine kinase (mtCK) abundance, unchanged cytosolic creatine kinase (CKM) levels, and significantly increased fetal creatine kinase (CKB) amounts (Figure 5a). While CKM is associated with the M-line of sarcomeres, CKB is predominantly found in the cytosol [40]. Visualization of CKM’s and CKB’s subcellular localization in relation to actin filaments showed a striated pattern of both CKM and CKB (Figure 5b,f), however, in comparison to the wild-type, the intensity profiles of both creatine kinase isoforms exhibited a phase-shift along the longitudinal axis of the myofibrils in the desmin knock-out cardiomyocytes (Figure 5c,g). For CKM, but not for CKB, this observation is also mirrored by a significant increase in the phase shift as quantitated by the degree of anisotropy algorithm (Figure 5d,h) in conjunction with significant decreases in the rate of both CKM and CKB colocalization with actin filaments (Figure 5e,i).

### 2.4. Proteome Analysis of Left Ventricular Cardiac Tissue Reveals Widespread Alterations Related to Subcellular Compartments and Metabolism

To delineate and quantitate changes in the protein expression inflicted by the lack of desmin, we performed quantitative mass spectrometry of left ventricular cardiac tissue from five animals of each genotype. In the homozygous desmin knock-out mice and their wild-type siblings, a proteome of 1522 protein groups was acquired at 1% FDR rate (raw data and a method description have been deposited to the ProteomeXchange Consortium via the PRIDE partner repository [41] (https://www.ebi.ac.uk/pride) with the dataset identifier PXD030938). Principal component analysis (PCA) depicted clear differences in the overall genotype-dependent protein expression profiles with desmin, as expected, as a major separator. Hierarchical clustering of all quantitated proteins created six clusters of Gene Ontology (GO) terms and Kyoto Encyclopedia of Genes and Genomes (KEGG) pathway enrichments (Figure 6a, Appendix A). While the upper three clusters (1), (2) and (3) denoted higher protein levels, the lower three clusters (4), (5) and (6) showed reduced protein expression in the desmin knock-out mice. Most prominent upregulation in desmin knock-out mice in cluster 1 referred to the extracellular compartment comprising terms annotated as “extracellular space”, “extracellular region”, “extracellular matrix”, “extracellular vesicle”, and “proteoglycans” (Appendix A). Similarly, cluster 2 referred to terms annotated as “sarcomere”, “M/A/I band”, “Z-disc”, “intercalated disc”, “focal adhesion”, “proteasome”, “mitochondrion”, “pyruvate metabolism”, “carbon metabolism”, “biosynthesis of amino acids”, “glycolysis and gluconeogenesis”, and “fructose and mannose metabolism” (Appendix A). Upregulation in cluster 3 comprised terms such as “cytoplasm”, “endoplasmic reticulum”, “protein processing in endoplasmic reticulum”, “sarcolemma”, “cell projection”, and “actin cytoskeleton” (Appendix A). Notably, most prominent downregulation in desmin knock-out mice in cluster 4 referred to terms comprising “mitochondrial membrane”, “pyruvate dehydrogenase complex”, “respiratory chain”, “ATP synthase complex”, “biosynthesis of amino acids”, “fatty acid degradation”, “fatty acid elongation”, “citrate cycle”, “oxidative phosphorylation”, “alanine, aspartate and glutamate metabolism”, “valine, leucine and isoleucine degradation”, and “glycolysis and gluconeogenesis” (Appendix A). A marked downregulation of different metabolic pathways is further highlighted in cluster 5, comprising terms such as “Z-disc”, “intercalated disc”, “myosin filament”, “desmosome”, “mitochondrial matrix”, “respiratory chain”, “biosynthesis of amino acids”, “citrate cycle”, “glycolysis and gluconeogenesis”, and “oxidative phosphorylation” (Appendix A). Finally, cluster 6 highlighted downregulation of terms such as “extracellular matrix”, “extracellular vesicle”, “caveola”, “focal adhesion”, “cardiac muscle contraction”, “mitochondrial membrane”, “respiratory chain”, “carbon metabolism”, “oxidative phosphorylation”, and “biosynthesis of amino acids” (Appendix A). Taken together, the hierarchical clustering approach and the subsequent GO term and KEGG pathway analyses further substantiated a broad range of metabolic derangements in left ventricular cardiac tissue of desmin knock-out mice. Notably, in addition to altered energy, fatty acid, and glucose metabolism, our analysis revealed dysregulated amino acid metabolism. Furthermore, marked imbalances in sarcomeric and extrasarcomeric cytoskeleton, protein homeostasis, as well as extracellular space-related processes and structures were observed. A group of related proteins including Xin actin-binding repeat-containing proteins 1 (Xirp1) and 2 (Xirp2), phosphoglucomutase-like protein 5 (Pgm5), nebulin-related-anchoring protein (Nrap) and kelch-like protein 41 (Klhl41), that are localized at intercalated discs in the normal heart, were all upregulated in desmin knock-out mice. These proteins are markers of striated muscle damage [42,43] or play a role in myofibril assembly and protection against myofibrillar damage [44,45,46,47]. Their increased expression in desmin knock-out left ventricular cardiac tissue confirms that a lack of desmin is associated with defects in the maintenance of the structural integrity of sarcomeres.

### 2.5. The Pattern of Significantly Regulated Proteins in Desmin Knock-Out Cardiac Tissue

Detailed quantitative analysis of individual proteins showed that a total of 98 proteins were significantly upregulated (*p* ≤ 0.05) in desmin knock-out mice when compared to their wild-type siblings. Of those, 43 were still significant after multiple testing *p*-value correction. A significant downregulation was detected for 97 proteins out of which 38 remained significant after *p*-value correction (Appendix A). A subset of significantly (*p* ≤ 0.05 and *p* ≤ 0.001, shown as negative decadic logarithm of the *p*-value) and markedly (fold change ≥2, shown as binary logarithm of the fold change) upregulated and downregulated proteins is visualized as a volcano plot (Figure 6b and Appendix A). One group of upregulated proteins was related to redox reactions and oxidative stress, and comprised glutathione peroxidase 3 (Gpx3), thioredoxin (Txn), thioredoxin domain-containing protein 5 (Txndc5), and protein disulfide-isomerase A4 (Pdia4). In keeping with the above mentioned upregulation of the term “extracellular matrix”, a second group consisting of vimentin (Vim), decorin (Dcn), and lumican (Lum) was significantly upregulated thus reflecting the previously and consistently documented increased interstitial fibrosis of left ventricular cardiac tissue in desmin knock-out mice [48,49,50]. In addition to the expected lack of desmin, two metabolism-related proteins, namely fructose-1,6-bisphosphatase isozyme 2 (Fbp2) and mitochondrial methionine-R-sulfoxide reductase B2 (Msrb2), were downregulated. For information on the other up- or downregulated proteins illustrated in the volcano plot (Figure 6b) please refer to the tabular overview (Appendix A).

### 2.6. A Closer Look at Metabolism-Related Proteins in Desmin Knock-Out Cardiac Tissue

Next, we focused specifically on metabolism-related proteins that were significantly up- or downregulated (*p* ≤ 0.05), with only a moderate fold change (1.1 ≤ fc ≤ 2.0) (Appendix A). The analysis revealed upregulation of only three relevant proteins in left ventricular cardiac tissue of desmin knock-out mice, namely UTP-glucose-1-phosphate uridylyltransferase (Ugp2, *p* = 0.0012, fc = 1.7), NADH-cytochrome b5 reductase 3 (Cyb5r3, *p* = 0.0092, fc = 1.6), and hexokinase-1 (Hk1, *p* = 0.0065, fc = 1.4). When focusing on downregulated metabolism-related proteins in the desmin knock-out genotype, findings were more complex. One group of downregulated proteins comprised core components of, and proteins related to, electron transport and oxidative phosphorylation mainly affecting complexes I, II, and V with *p*-values in a range from 0.049 to 0.00022 and fold changes between 1.15 and 1.59 (Adck3 (coenzyme Q biosynthesis), Atp5a1, Atp5b, Atp5i/Atp5k, Atp5j, Atp5o, Cox5b, Cycs, Mtatp8, Ndufa12, Ndufa13, Ndufa6, Ndufs2, Ndufs3, Ndufv2, Sdhb, and Uqcrc2; Appendix A). In addition, the amount of electron transfer flavoprotein subunit beta (Etfb) was significantly decreased (*p* = 0.00015, fc = 1.42; Appendix A). A second group contained proteins of the citrate cycle with *p*-values ranging from 0.043 to 0.0028 and fold changes between 1.21 and 1.72 (Aco2, Cs, Dld, Fh, Idh2, Mdh2, Pdha1, Suclg1, and Suclg2; Appendix A). The third group consisted of several enzymes involved in the mitochondrial beta-oxidation of fatty acids with *p*-values ranging from 0.046 to 0.00011 and fold changes between 1.14 and 1.85 (Acaa2, Acadm, Acads, Acadvl, Acat1, Decr1, Ech1, Echs1, Eci1, Eci2, Hadh, Hadha, Hadhb, Hsd17b10/Hadh2, Mut, Pcca, and Pccb; Appendix A). Notably, these downregulated enzymes were related to the metabolism of short, medium, and long-chain, as well as unsaturated, branched-chain, straight, and odd fatty acids. Also, four proteins involved in the transport, activation, and availability of fatty acids, namely carnitine O-acetyltransferase (Crat, *p* = 0.000028, fc = 1.50) and carnitine O-palmitoyltransferase 2 (Cpt2, *p* = 0.0033, fc = 1.32) as well as the acyl-CoA hydrolysing enzymes Acot1 and Acot2 (*p* = 0.0062, fc = 1.52), were decreased in desmin knock-out cardiac tissue (Appendix A). A further downregulated protein was acyl-CoA synthetase short chain family member 1 (Acss1, *p* = 0.00047, fc = 1.44) (Appendix A). The fifth group comprised proteins exerting functions in amino acid metabolism, mostly associated with degradation, with *p*-values in a range from 0.041 to 0.00031 and fold changes between 1.21 and 1.52 (Aldh5a1, Aldh6a1, Auh, Bckdha, Bckdhb, Dld, D10Jhu81e, Got1, Hsd17b10/Hadh2, Mccc1, Mccc2, Pcca, and Pccb; Appendix A). A single protein was involved in glycogen degradation (Agl, *p* = 0.033, fc = 1.26; Appendix A). A sixth group with *p*-values ranging from 0.042 to 0.00013 and fold changes between 1.14 and 1.50 consisted of one protein related to mitochondrial fission (Mtfp1) and two proteins related to mitophagy (Phb and Phb2; Appendix A). Finally, a seventh group of downregulated proteins with *p*-values in a range from 0.049 to 0.0054 and fold changes between 1.17 and 1.50 was related to redox reactions and oxidative stress (Gstk1, Gstm1, Prdx5, Sod1 (cytosolic), and Sod2 (mitochondrial); Appendix A).

## 3. Discussion

The lack of desmin in human hearts triggers a cascade of noxious cellular effects often leading to progressive heart failure during adolescence or early adulthood [14]. Findings in patients and in desmin knock-out mice indicate that the key process in the molecular pathogenesis of this very rare desminopathy subform is generalized destabilization of the extrasarcomeric cytoskeleton due to the lack of its major component [37,51]. Subsequent alterations in the structural and functional organization of the extrasarcomeric cytoskeleton negatively interfere with the ordered alignment and proper subcellular attachment of the entire myofibrillar apparatus, which likely explains impaired muscular force generation and impaired mechanical stress resistance [21]. A second, major disease promoting factor appears to be the negative impact of desmin deficiency on myocardial metabolism. Already the first analyses of desmin knock-out mice showed that the extrasarcomeric desmin cytoskeleton is tightly linked to subcellular distribution and respiratory function of mitochondria in striated muscle cells [20,31]. The importance of this interplay is supported by the observation that aside from *DES* mutations [30,52], mutations in genes encoding other essential extrasarcomeric intermediate filament cytoskeleton components such as the cytoskeletal linker protein plectin (*PLEC*) [53,54] or the small heat shock protein alphaB-crystallin (*CRYAB*) [55,56] also cause cardiomyopathies and myopathies with morphological and biochemical evidence of mitochondrial dysfunction [30,57,58]. In addition to direct interactions between the N-terminal desmin domain with the outer mitochondrial membrane [59], plectin isoform 1b has been identified as an important factor concerting associations between mitochondria and the three-dimensional desmin network in striated muscle cells [60]. While recent studies addressed the general molecular crosstalk between mitochondria and the cytoskeleton in striated muscle cells (for review see [61,62]), insight into the specific metabolic homeostasis of diseased cardiac tissue harboring *DES, PLEC,* or *CRYAB* mutations is currently lacking. To delineate general metabolic effects associated with the lack of desmin, here, we performed a comprehensive morphological, biochemical, and proteomic analysis of left ventricular cardiac tissue derived from six-month-old desmin knock-out mice and their wild-type siblings kept under standard ‘sedentary’ housing conditions.

### 3.1. Desmin and Myocardial Mitochondria

When we analyzed three-dimensional mitochondrial distribution by means of VDAC1-stained left ventricular sections, we observed rarefied mitochondrial networks in desmin knock-out mice. This finding implied a significant reduction in mitochondrial number. Indeed, enzymatic measurements showed a tendency towards lower malate dehydrogenase and citrate synthase activities, and quantitative real time PCR analysis yielded significantly reduced mtDNA copy numbers. The latter finding resembles previous results obtained in skeletal muscle specimens derived from patients with desmin missense mutations as well as from homozygous desmin knock-out and homozygous R349P desmin knock-in mice [30]. In contrast, transgenic mice expressing a desmin variant harboring a 7-amino acid deletion displayed an increased mtDNA copy number in cardiac tissue. This study further implicated that increased mitochondrial content may have been related to an imbalance in mitochondrial fission and fusion processes [63]. In this respect, our proteomic data revealed a significant downregulation of mitochondrial fission process protein 1 (Mtfp1), however, no changes in Mfn1 and Opa1, and significant downregulation of prohibitin proteins (Phb, Phb2) that are involved in mitophagy by targeting mitochondria for autophagic degradation [64]. Additionally, our proteomic analysis depicted three downregulated mitochondrial proteins implicated in redox reactions and oxidative stress, namely methionine-R-sulfoxide reductase B2 (Msrb2), peroxiredoxin-5 (Prdx5), and manganese superoxide dismutase (Sod2).

### 3.2. Desmin and the Ultrastructure and Fragility of Mitochondria in Cardiac Muscle Tissue

Our ultrastructural analysis revealed areas of focal mitochondrial clustering, as well as a markedly coarsened mitochondrial cristae in cardiac tissue of desmin knock-out mice. Both findings mirror the previously described mitochondrial pathology in the hearts of these mice [31,65]. We also noted the previously described focal clustering of mitochondria in desmin deficient cardiomyocytes [31]. Another characteristic finding of our ultrastructural analysis was the presence of electron-dense lipofuscin deposits, that were much more abundant in the desmin knock-out cardiomyocytes. In addition to the ultrastructural alterations, our attempts to enrich mitochondrial fractions failed due to the apparently high fragility and swelling of the extracted mitochondria from the desmin knock-out cardiac tissue. The latter findings indicate structural changes of mitochondria in the desmin knock-out genotype, which negatively impact their mechanical stability during the extraction or fractionation process.

### 3.3. From Aberrant Mitochondria to Changes in Fatty Acid Metabolism and Oxidative Phosphorylation

Numerical and structural mitochondrial alterations prompted us to analyze mitochondrial fatty acid metabolism, which is the major ATP generator in cardiac muscle [66]. In addition to significantly decreased fatty acid transporter CD36 expression, octanoyl-carnitine-stimulated beta-oxidation showed a significant reduction in desmin knock-out cardiac tissue homogenates. Notably, dried whole blood sample analysis depicted an intriguing picture in which multiple acylcarnitines ranging from C3 to C18 chain length showed significantly increased concentrations in desmin knock-out mice. Regarding this finding, one has to keep in mind that acylcarnitine blood levels are determined to a far greater extent by skeletal than the cardiac muscle metabolism. However, proteomic analysis of the left ventricular cardiac tissue also depicted that a spectrum of proteins involved in fatty acid metabolism to oxidative phosphorylation was significantly downregulated. This included proteins involved in the transport and activation of fatty acids such as the carnitine O-palmitoyltransferase 2 (Cpt2), which is active with medium and long-chain acyl-CoA esters for subsequent beta-oxidation, as well as multiple core enzymes involved in mitochondrial beta-oxidation, the citrate cycle, electron transport, and oxidative phosphorylation. Taken together, our enzymatic and mass spectrometric analyses documented widespread aberrations of fatty acid metabolism and oxidative phosphorylation in our desmin knock-out mice.

### 3.4. Metabolic Adaptations in Desmin Knock-Out Hearts

Key findings regarding glucose metabolism in desmin knock-out left ventricular cardiac tissue were a markedly upregulated fetal GLUT1 isoform in conjunction with GLUT4 with higher apparent molecular weight in immunoblotting, and increased hexokinase isoform 1 protein expression. Furthermore, hexokinase enzymatic activity (assay not isoform specific) was significantly increased. GLUT4′s shifted apparent molecular weight is consistent with posttranslational modifications, e.g., phosphorylation or N-glycosylation [67]. Moreover, GLUT4′s subsarcolemmal enrichment in desmin knock-out cardiomyocytes may result from such posttranslational modifications [68]. The observed dissociation from the sarcolemma of both GLUT1 and GLUT4 in desmin knock-out cardiomyocytes may be linked to the loss of t-tubular system (a target structure of GLUT4 containing vesicles) and accumulation of the GLUT4-containing sarcoplasmic vesicles underneath the sarcolemma [69]. The marked upregulation of GLUT1 expression, which is under control of SP1 transcription factor in embryonal and neonatal heart [70], suggests activation of fetal gene program and increased glucose metabolism in the desmin knock-out heart. Proteomic analysis also revealed a significant upregulation of UTP-glucose-1-phosphate uridylyltransferase (Ugp2). The enzyme catalyzes glucose-1-phosphate conversion to UDP-glucose. The gluconeogenesis regulatory fructose-1,6-bisphosphatase isozyme 2 (Fbp2) and glycogen debranching enzyme Agl were downregulated. The outlined findings imply an increase in glucose metabolism or utilization to compensate for decreased mitochondrial ATP-generation. Moreover, the creatine kinase system exhibited a significant reduction in mitochondrial creatine kinase (mtCK) amounts and significantly increased cytosolic fetal creatine kinase (CKB). CKB is more resistant to inactivation by reactive oxygen species and possesses a higher affinity to phosphocreatine than CKM [71]. Thus, the enzyme can operate under conditions of increased oxidative stress and lower phosphocreatine concentration. The observed phase-shift of both CKM and CKB in the desmin knock-out myofibrils towards the levels of sarcomeric M-lines suggests enhanced metabolic channeling of ATP to myosin-ATPase by the sarcomeric creatine kinase system [72,73]. The response likely compensates for the impairment of intermyofibrillar mitochondria. In light of the decreased mitochondrial stability, which is largely dependent on mtCK octamers [74] and prohibitins [75,76], it is noteworthy that mtCK, Phb, and Phb2 were downregulated in the desmin knock-out cardiomyocytes.

In contrast to our proteomic analysis using total left ventricular cardiac tissue, a previous study focused on the analysis of mitochondria that were fractionated from total heart tissue [4]. Due to the sample preparation and proteomic measurement methodologies, i.e., two-dimensional gel electrophoresis in conjunction with MALDI-TOF mass spectrometry, these results are only partially comparable with our present work. For example, various citrate cycle proteins were unchanged. Further examples are the mixed patterns of non-, up- or downregulated proteins of amino acid metabolism, respiratory chain, oxidative phosphorylation, glucose metabolism and oxidative stress [4]. Another study reported the absence of kinesin in heart mitochondria fractions and in the heart tissue of desmin knock-out mice using immunoblotting and immuno-electron microscopy, respectively [5], a finding that could not be recapitulated in our analysis showing no significant change in the amount of kinesin-1 heavy chain (Kif5b). Corresponding data derived from patients with a desmin knock-out cardiomyopathy have not yet been published. In the context of human desminopathies it is, however, noteworthy that another study reported a marked decrease in the levels of respiratory chain proteins as well as in citrate synthase activity in cardiac tissue of patients harboring heterozygous *DES* mutations [52].

### 3.5. Desmin Deficiency and the Heart: A Combined Structural and Metabolic Disease

Desmin knock-out cardiomyopathy is considered a ‘structural cardiomyopathy’ caused by deficiency in an essential component of the extrasarcomeric cytoskeleton. However, our data suggest that desmin deficiency is also associated with profound abnormalities in myocardial metabolism, compatible with a secondary ‘mitochondrial cardiomyopathy’. The starting point of the metabolic mayhem seems to be directly related to a defective and stressed mitochondrial compartment. Reduced mitochondrial content along with the structural aberrations provide an explanation for compromised fatty acid metabolism and oxidative phosphorylation, which most likely results in reduced ATP generation in the very energy demanding cardiac tissue. As an apparent countermeasure, glucose metabolism and fetal creatine kinase isoform (CKB) were increased. We therefore postulate that the cardiomyopathy associated with desmin deficiency should be conceptualized as combined structural and mitochondrial cardiomyopathy.

Increased blood concentrations of short, intermediate, and long acylcarnitines associated with elevated blood levels of branched-chain amino acids also occur in the human Multiple Acyl-CoA Dehydrogenase Deficiency (MADD) syndrome (MIM #231680; [77]). MADD, which can cause cardiomyopathy [78], is either attributed to mutations in the genes coding for the mitochondrial electron transfer flavoprotein-ubiquinone oxidoreductase (*ETFDH*), electron transfer flavoprotein subunit alpha (*ETFA*), or electron transfer flavoprotein subunit beta (*ETFB*) (MIM #231680). While ETFDH and ETFA were unchanged in the cardiac tissue proteome from our desmin knock-out mice, the amount of ETFB was significantly decreased (*p* = 0.00015, fc = 1.42; Appendix A). Furthermore, combined increases in 3OH-butyryl-carnitine (C4OH) and branched-chain amino acid blood levels indicate a catabolic metabolic state. Human desminopathies with a lack of desmin protein are very rare and, to our knowledge, no published data on acylcarnitines and amino acid blood levels are currently available. Review of clinical chemistry data in a previously reported patient (patient 2 in [14]) showed no blood acylcarnitine elevations. However, this patient showed an acyl CoA dehydrogenase deficiency-characteristic increase in the C8/C10-carnitine ratio [79] with a value of 0.98 (normal range, 0.60 to 0.00), a finding that was also present in our desmin knock-out mice (2.12 vs. 1.85, *p* = 0.049; Table 1). Another enzyme of interest that was also significantly decreased in our proteomic analysis is 2,4-dienoyl-CoA reductase (Decr1; *p* = 0.046, fc = 1.24; Appendix A). This enzyme is essentially involved in the pathogenesis of 2,4-dienoyl-CoA reductase deficiency (DECRD), a disease with a characteristic increase in 2-trans 4-cis-decadienoyl-carnitine (C10:2) blood levels [80]. In line with increased circulating acylcarnitine and amino acid concentrations, carnitine acyltransferases, acyl-CoA hydrolysing enzymes, proteins of the mitochondrial beta-oxidation, and the amounts of enzymes with predominantly catabolic functions in the amino acid metabolism were significantly decreased in desmin-deficient left ventricles.

### 3.6. Translational Aspects Derived from the Analysis of Murine Desmin Knock-Out Hearts

Severe metabolic derangements seem to be common in various human cardiomyopathies. Specifically, a recent multi-omics study focusing on human hypertrophic cardiomyopathy reported reduced mitochondrial cristae densities, reduced mtDNA copy numbers, and multiple mitochondrial metabolic derangements comprising reduced oxidative respiration, decreased levels of citrate cycle intermediates and high energy phosphate metabolites, and the accumulation of free fatty acids [81]. These mitochondrial changes are at least partly mirrored in our desmin knock-out animals, thus indicating that dysfunctional mitochondria are a common feature in cardiomyopathies irrespective of the underlying aetiology. However, human hypertrophic cardiomyopathy tissues exhibited significantly reduced acylcarnitines, whereas the analysis of blood acylcarnitines in our desmin knock-out mice showed an opposite picture. To address such discrepancies, further studies are required.

In the context of widespread mitochondrial dysfunction in desmin knock-out hearts, it may be prudent to avoid medications that worsen mitochondrial function [82] in patients lacking desmin. Whether other metabolic stressors, such as physical exertion, pose risks deserves to be studied. A high mortality rate has been observed in desmin knock-out mice exposed to a forced swimming exercise protocol [83,84], which might be due to an acute metabolic crisis. On the other hand, moderate physical exercise could stimulate mitochondriogenesis and improve oxidative metabolism in the long run [82]. Possibly, therapeutic interventions for mitochondriopathies [82] may also be beneficial in patients suffering from desminopathy-associated heart diseases. For example, as treatment of the Multiple Acyl-CoA Dehydrogenase Deficiency syndrome with riboflavin [85,86], riboflavin together with coenzyme Q10 [87], riboflavin together with carnitine [88], or coenzyme Q10 [88] may markedly improve the muscle symptoms, it is tempting to speculate that such a therapy may also improve the metabolic condition in desmin deficient hearts.

## 4. Materials and Methods

### 4.1. Animals

We studied six-month-old homozygous desmin knock-out mice B6J.129S2/Sv-*Des*^tm1Cba^/Cscl (http://www.informatics.jax.org/allele/MGI:2159584; [20]) and their wild-type siblings. Mice of both sexes were used in approximately similar numbers. Routine PCR genotyping was performed using primers DES 1 5’-TTGGGGTCGCTGCGGTCTAGCC-3’, DES 1R 5’-GGTCGTCTATCAGGTTGTCACG-3’, and LacZ 430R 5’-GATCGATCTCGCCATACAGCGC-3’ resulting in products of 350 bp for the wild-type and 450 bp for the knock-out allele. In addition, the absence of desmin was verified by immunoblotting in individual animals. Mice were housed in isolated ventilated cages (IVC) under specific and opportunistic pathogen-free (SOPF) conditions in a standard environment with free access to water and food. Health monitoring was done as recommended by the Federation of European Laboratory Animal Science Associations (FELASA). Mice were handled in accordance with the German Animal Welfare Act (Tierschutzgesetz) as well as the German Regulation for the protection of animals used for experimental or other scientific purposes (Tierschutz-Versuchstierverordnung). For tissue dissection, mice were euthanized by cervical dislocation. All investigations were approved by the governmental office for animal care (Landesamt für Natur, Umwelt und Verbraucherschutz North Rhine-Westphalia (LANUV NRW), Recklinghausen, Germany (reference numbers 84-02.04.2014.A262 and 84-02.05.40.14.057)).

### 4.2. Patients

Clinical chemistry data derived from a patient, who has been included in a previous publication (patient 2, [14]), were reviewed and included in this study. Blood samples from the patient were obtained upon written informed consent according to the Declaration of Helsinki and approval by the Boğaziçi University Institutional Review Board for Research with Human Participants (reference number 20922).

### 4.3. Mass Spectrometric Analysis of Acylcarnitine and Amino Acid Levels in Blood

For mass spectrometric quantitation of acylcarnitines and amino acids, retro-orbital sinus blood samples from mice under isoflurane anesthesia (single administration of a dose of 4% Forane via inhalation in a small chamber) were withdrawn using Pasteur pipettes immediately prior to euthanasia by cervical dislocation for muscle tissue dissection and collected on dried blood sample cards.

Acylcarnitines were extracted from discs of 3 mm diameter, which were punched out of the dried blood sample cards, with an acetonitrile/water-based buffer (NEO Extraction Buffer #55008, Chromsystems, München, Germany) containing isotope-labelled acylcarnitines as internal standard (C0-carnitine-D9, C2-carnitine-D3, C3-carnitine-D3, C4-carnitine-D3, C5-carnitine-D9, C5DC-carnitine-D6, C6-carnitine-D3, C8-carnitine-D3, C10-carnitine-D3, C12-carnitine-D3, C14-carnitine-D3, C16-carnitine-D3, C18-carnitine-D3). After centrifugation at 14,000× *g* for 5 min the supernatants were evaporated to dryness at 60 °C in a stream of nitrogen and butylated by addition of anhydrous butanol/HCl. After drying and reconstitution of the acylcarnitines in an acetonitrile/water-based buffer (NEO Reconstitution Buffer #55006, Chromsystems), samples were directly injected into the ESI source of an LC-MS/MS instrument (Quattro premier XE mass spectrometer, Waters, Milford, MA, USA) and analyzed in Parent-Ion-Scan (PIS) mode with the MS1 scan range set from 200 to 500 m/z. A characteristic 85 m/z fragment ion generated from acylcarnitine precursor ions selected for collision-induced dissociation fragmentation was selected by MS2 and detected by a photo multiplier. Primary data analysis was performed using MassLynx 4.1 with the NeoLynx tool (Waters, USA).

Amino acids were extracted from another 3.2 mm diameter discs punched out of the dried blood sample cards into Eppendorf reaction vials, and 50 µL water were added to each sample. Prior to analysis, all plasma calibration standards and control samples were thawed and allowed to equilibrate at room temperature; 50 µL of each were transferred into Eppendorf reaction vials. Extraction was carried out with 200 µL methanol containing isotope-labelled amino acids (cell-free amino acid mixture 13C, 15N; Sigma Aldrich, Merck, Darmstadt, Germany; 500 µL in 200 mL methanol). After agitation at 1000 rpm for 20 min at 20 to 25 °C all vials were centrifuged at 16,000× *g* for 5 min. For derivatization, 10 µL of the supernatants were transferred into sample cups and mixed with 70 µL of borate buffer and 20 µL derivatization reagent 6-aminoquinolyl-N-hydroxysuccinimidyl carbamate in acetonitrile (included in AccQ-Tag Ultra Derivatization Kit, Waters, Eschborn, Germany). All vials were incubated for 10 min at ambient temperature. Chromatographic separation of partly isobaric compounds was carried out on a ACQUITY UPLC I-Class System with a CortecsUPLC column (particle size 1.8 µm, 150 mm length, 2.1 mm inner diameter) (Waters, Germany) using 0.1% formic acid in ULC water and 0.1% formic acid in acetonitrile as mobile phase. After chromatographic separation, detection was performed using a Xevo TQS-micro (Waters, Germany) in ESI positive mode quantification with MassLynx NT version 4.1 (Waters, Germany). A decrease in acylcarnitine and amino acid amounts on the dried blood sample cards during storage for a few months [89] was considered.

### 4.4. Cardiac Muscle Tissue Preparation

Murine hearts were dissected and processed for different subsequent analyses. For quality control, the genotype of all dissected animals was confirmed by a second PCR. Explanted hearts were washed in ice-cold saline solution and left ventricles (LV) were separated from the free wall right ventricle and immediately frozen in liquid nitrogen. Frozen tissue specimens were pulverized in liquid nitrogen and homogenized in homogenization buffer (12.5 mM Tris, 2.5 mM EGTA, 1 mM EDTA, 250 mM sucrose, 5 mM DTT, Complete protease inhibitor cocktail (Roche Diagnostics, Mannheim, Germany ), pH 7.4). Total protein concentrations were assessed using the Bradford Method Protein Assay Kit (Sigma Aldrich). The homogenates were aliquoted and stored at −80 °C until they were used for western blotting (WB) and enzyme activity assays. For immunofluorescence analysis, the left ventricles of a set of explanted hearts were immediately perfused by injection of relaxing Tyrode solution (140 mM NaCl, 5.4 mM KCl, 1 mM Na_2_HPO_4_, 1 mM MgCl_2_.6H_2_O, 10 mM glucose, 5 mM HEPES, pH 7.4) and subsequently perfused with freshly prepared 4% formaldehyde (Sigma Aldrich) solution. Perfused hearts were transferred into fresh fixative and kept immersed for 2 h. Subsequently, they were infiltrated with 20% cryoprotective sucrose solution, snap-frozen in liquid nitrogen, and stored at −80 °C. For mtDNA and proteomic analyses another set of explanted hearts was used for separation of the cardiac apexes and left ventricles, which were snap-frozen in liquid nitrogen. For ultrastructural analysis, several cardiac apexes were fixed in glutaraldehyde.

### 4.5. Enzyme Activity Measurements

Specific enzyme activities (U/mg protein) of hexokinase (HK), phosphofructokinase (PFK), malate dehydrogenase (MDH), and citrate synthase (CS) were spectrophotometrically assessed by enzyme coupled assays using either a 96-well multi-reader system BioTek Synergy HT (HK, PFK; Agilent, Santa Clara, CA, USA) or a spectrophotometer Shimadzu-UV1601 (MDH, CS; Shimadzu corporation, Duisburg, Germany).

The enzyme activity of HK was assessed as described by an enzyme-coupled assay using a slightly modified Worthington protocol (http://www.worthington-biochem.com/HK/assay.html (accessed on 5 October 2022)) [39]. Briefly, 60 μg of the protein samples were loaded onto a 96-well plate and diluted in assay buffer (50 mM Tris, 13.3 mM MgCl_2_, 0.8 mM NAD^+^, 0.8 mM ATP, 1 U/mL glucose-6-phosphate dehydrogenase (Roche/Sigma-Aldrich/Merck), pH 8.0). The reaction was initiated after 2 min incubation at 30 °C by the addition of a solution containing 1.5 M glucose, 50 mM Tris, 13.3 mM MgCl_2_, pH 8.0. The absorbance was recorded for 15 min at 339 nm.

The enzyme activity of PFK was assessed using the Activity Colorimetric Assay Kit (MAK093, Sigma-Aldrich) according to the manufacturer’s instructions. For this purpose, 20 mg pulverized tissue samples were homogenized in 200 μl ice-cold PFK assay buffer and samples were centrifuged at 13,000× *g* for 10 min to remove insoluble material. The supernatants were loaded into the assay in a volume of 10 µL and mixed with the reaction buffer containing fructose-6-phosphate and ATP. The resulting colorimetric product was measured at 450 nm. Values are proportional to the PFK activity.

The MDH specific enzyme activity was measured using a Worthington protocol that determines the decrease in absorbance at 340 nm resulting from the oxidation of NADH (http://www.worthington-biochem.com/MDH/assay.html (accessed on 5 October 2022)). 20 µg of protein was resuspended in 100 mM phosphate buffer, pH 7.4, containing 0.30 mM NADH. The reaction was initiated by addition of 1.51 mM oxaloacetate.

The specific enzyme activity of CS was detected at 412 nm using the 5,5’-dithio-bis(2-nitrobenzoic acid) (DTNB) method. DTNB reacts with the free sulfhydryl group of coenzyme A and produces mercaptide ions. The reaction was performed in 1 mL of buffer containing 100 mM Tris (pH 8.1), 0.1 mM DTNB and 0.12 mM acetyl-coenzyme A, and was started by adding 0.5 mM oxaloacetate as described previously [90].

### 4.6. High-Resolution Respirometry

Respiration of left ventricle cardiac tissue homogenates from desmin knock-out mice and wild-type littermates was determined by using, as described previously, [90] high-resolution respirometry (Oxygraph 2-k, Oroboros) using MiR05 (0.5 mM EGTA, 3 mM MgCl_2_, 60 mM K^+^-lactobionate, 20 mM taurine, 10 mM KH_2_PO_4_, 20 mM HEPES, 110 mM D-sucrose) and 1 g/l BSA as respiration medium [91,92]. To analyze the mitochondrial beta-oxidation rate, the following substrates were added: 0.5 mM malate, 2.5 mM ADP, 0.25 mM octanoyl-carnitine, and 0.01 mM cytochrome *c*.

### 4.7. SDS-PAGE and Immunoblotting

Left ventricle cardiac tissue homogenates (each 20 µg protein) were separated by SDS-PAGE as described previously [93] using 10–12% polyacrylamide gels and a voltage gradient ranging from 100 to 150 V (Mini-PROTEAN TetraCell, Bio-Rad, Hercules, California, USA). Separated proteins were electro-transferred onto a nitrocellulose membrane (0.2 μm pore size, Protran BA 83, Whatman, Merck) at a constant voltage of 25 V for 35 min (Trans-Blot Turbo, Bio-Rad). Membranes were washed, stained by Ponceau S solution (Sigma), and scanned for further evaluation. De-stained membranes were blocked for 1 h at room temperature with 5% non-fat dry milk powder in Tris-buffered saline solution (20 mM Tris, 500 mM NaCl, pH 7.5) containing 0.05% Tween 20 (TBS-T). After washing in TBS-T, membranes were incubated overnight at 4°C with the following primary antibodies: rabbit polyclonal anti-HK1 and anti-HK2 (Abcam, Cambridge, UK; ab150423, 1:2000, and ab78259, 1:500, respectively,), rabbit polyclonal GLUT1 (ab115730, 1:1000), mouse monoclonal GLUT4 (ab65267, 1:1000), rabbit polyclonal anti-CD36 (ab64014, 1:500), and goat polyclonal anti-CKB, CKM or mtCK (Santa Cruz Biotechnologies, Dallas, Texas, USA; sc-15157, 1:500, sc-15164, 1:1000, and sc-15168, 1:400, respectively). After washing in TBS-T, the membranes were incubated with HRP-conjugated anti-rabbit (A9169, Sigma Aldrich, Merck), anti-mouse (Thermo Fisher Scientific, Waltham, MA, USA; 32430, 1:10,000) or anti-goat (Santa Cruz Biotechnologies, sc-2033, 1:10,000) secondary antibodies for 1 h. After final washing, the SuperSignal West Dura Extended Duration Substrate (Thermo Fisher Scientific) was added and the chemiluminescence signals were visualized using a LAS-3000 Imaging System from Fuji (Biocompare, South San Francisco, CA, USA). Densitometric quantitation of specific protein bands was performed using ImageJ. All samples from each group were always separated on a single gel. Western blotting data were normalized against total protein amounts that were visualized by Ponceau S staining.

### 4.8. Immunofluorescence Staining, Imaging, and Image Analysis

Using a Leica CM3050 cryostat (Leica Microsystems, Wetzlar, Germany), 5 to 7 μm-thick sections of the cryopreserved left ventricular cardiac tissue specimens were prepared. Cryosections were rehydrated in PBS, permeabilized in ice-cold methanol, briefly incubated in 1% SDS for antigen retrieval, and incubated in a blocking solution containing 10% donkey serum, 10% goat serum, 0.3% Triton X-100, and 0.3 M glycine in PBS. Methanol permeabilization and SDS treatment were omitted for sections dedicated to Wheat Germ Agglutinin (WGA) staining. The sections were incubated with primary antibodies (rabbit polyclonal anti-HK1 and anti-HK2 (Abcam, ab150423, 1:50, and ab78259, 1:50, respectively), rabbit polyclonal anti-Glut1 and anti-Glut4 (Abcam, ab115730, 1:50, and ab654, 1:50, respectively), and rabbit polyclonal anti-CKB and anti-CKM (Abcam, ab92452, 1:50, and ab189438, 1:50, respectively)) and conjugated secondary antibody (donkey anti-rabbit IgG AlexaFluor488 (Thermo Fisher Scientific, A-21206, 1:200)) followed by staining with a subcellular compartment marker (total OXPHOS Blue Native WB antibody cocktail (Abcam, ab110412, 1:200) and goat anti-mouse IgG AlexaFluor647 (Thermo Fisher Scientific, A-21235, 1:200) for mitochondria, WGA AlexaFluor647 conjugate (Thermo Fisher Scientific, W-32466, 1:200) for sarcolemma and t-tubules, and Phalloidin AlexaFluor647 conjugate (Thermo Fisher Scientific, A-22287, 1:40) for actin filaments). Sections were mounted in ProLong Gold Antifade Reagent with DAPI (Thermo Fisher Scientific).

Images were acquired using a wide-field inverted fluorescence microscope (Olympus IX2-UCB) equipped with a MT20 mercury arc illumination unit (Olympus, Tokyo, Japan), a fully motorized stage Corvus (PI-Japan, Kanagawa, Japan), and a CCD camera Orca C4742-80-12AG (Hamamatsu Photonics, Seoul, Korea). Samples were observed with a 100x 1.4NA Plan-Apochromat objective lens. Filter combinations were as follows: DAPI (blue), triple-band set 69002-ET-DAPI/FITC/TexasRed (Chroma Technology Corp., Vermont, USA), ex. 350 nm (bandwidth 50 nm), em. 457 nm (bandwidth 22 nm); proteins of interest (green), U-MWIBA3 (Olympus), ex. 477.5 nm (bandwidth 17.5 nm), em. 530 nm (bandwidth 20 nm); additional markers (red), U-N41008 (Chroma Technology Corp.), ex. 620 nm (bandwidth 60 nm), em. 700 nm (bandwidth 75 nm). For each sample, images were taken from at least 5 randomly selected positions on each sample. Each position was optically sectioned at 0.5 µm steps resulting in approximately 8–12 layers in a Z-stack depending on specimen thickness.

Subcellular colocalization of proteins of interest with mitochondria, sarcolemma, t-tubules, and actin filaments was expressed as Manders M1 coefficient [94], and calculated using the Colocalization Threshold plug-in in Fiji software [95]. Prior to colocalization analyses, images were calibrated, and a rolling ball background subtraction with radius (r = 5) was applied. To solve the relative disproportion in sarcolemma and t-tubules signal intensities, the Contrast Limited Adaptive Histogram Equalization (CLAHE) [96] was used to enhance the fluorescent signal of WGA AlexaFluor647 conjugate. For representative images, foreground pixels were enhanced using the Difference of Gaussian (DoG) algorithm [97]. The extent of the striation pattern of the CKM/CKB fluorescence signal was evaluated using the degree of anisotropy algorithm, a part of the BoneJ plug-in collection for Fiji software [98]. On the selected, representative images, qualitative evaluation of the phase-shift between the CKM/CKB and WGA fluorescence signal was performed by plotting a linear intensity profile alongside the longitudinal axis of the myofibrils.

### 4.9. Confocal Image Acquisition and Analysis

Cryosections of the left ventricular cardiac tissues were also stained with an antibody directed against the voltage-dependent anion channel (rabbit polyclonal anti-VDAC1, Abcam, ab15895, 1:100; secondary antibody goat anti-rabbit IgG AlexaFluor555, Thermo Fisher Scientific, A-21429, 1:100). Confocal Z-stacks were acquired on a Leica TCS SP5 microscope using a HCX PL APO CS 63.0 × 1.30 GLYC objective lens. To enhance the signal and remove noise and background, images were deconvolved using Huygens Professional (version 17.04; Scientific Volume Imaging, Amsterdam, The Netherlands). Further analysis was performed with Fiji [99]. Fiji’s Tubeness filter (σ = 0.15 µm) was applied to enhance filamentous structures in the image volumes. The mitochondrial network was then separated from the background using automatic Otsu thresholding. The midlines of the network were extracted by the Skeletonization algorithm, and the number of branch points was counted. The mitochondrial network was visualized using 3Dscript [99].

### 4.10. Ultrastructural Analysis

For transmission electron microscopy, cardiac apexes were fixed in freshly prepared 2.5% glutaraldehyde in 0.1 M Sørensen’s phosphate buffer, pH 7.2, with 0.23% NaCl, post-fixed in 2% buffered osmium tetroxide, dehydrated in graded ethanol concentrations, and embedded in epoxy resin. 1 µm-semi-thin-sections for orientation were stained with toluidine blue. Ultra-thin sections were stained with uranyl acetate and lead citrate and were examined with a LEO 906E transmission electron microscope (Carl Zeiss Microscopy GmbH, Oberkochen, Germany).

### 4.11. mtDNA Deletions Determination

Total DNA was isolated from the apexes of dissected hearts using the QIAamp DNA Mini Kit (QIAGEN, Venlo, The Netherlands) column purification. The DNA isolation protocol provided by the manufacturer was followed and the samples were eluted in 200 µL elution buffer provided in the kit. All samples were stored at 4 °C. Deletions of mitochondrial DNA were detected by a long-range PCR protocol using the TaKaRa LA Taq Hot Start polymerase (Takara, Saint-Germain-en-Laye, France) specifically suitable for the production of longer PCR amplicons with greater accuracy. In order to detect deletions, almost the entire mtDNA was amplified between primers musMT2482F24 (5′-GTTCAACGATTAAAGTCCTACGTG-3′) and musMT1005R24 (5′-CCAGTATGCTTACCTTGTTACGAC-3′), under the following conditions: 95 °C for 2.5 min, 30 cycles of 92 °C for 20 s and 66.8 °C for 5.5 min, and final extension at 72 °C for 10 min. The PCR products were visualized on a 1% agarose gel with Quick-Load 1 kb Extend DNA Ladder (New England Biolabs, Frankfurt/Main, Germany).

### 4.12. mtDNA Copy Number Determination

Total DNA, isolated as described above, was used for mtDNA copy number determination by quantitative real-time PCR (qPCR). The qPCR was performed with 2×SYBR Green qPCR Master Mix (Bimake, Munich, Germany). Three different DNA concentrations were used (1.6 ng/µL, 0.8 ng/µL, and 0.4 ng/µL) with final DNA amounts of 20 ng, 10 ng, and 5 ng, respectively, and each sample was used in triplicate for each dilution. Primers musMT553F23 (5′-GCCAGAGAACTACTAGCCATAGC-3′) and musMT668R23 (5′-AGCAAGAGATGGTGAGGTAGAGC-3′) were used for mtDNA amplification. Amplification of the single copy gene for the inward rectifier potassium channel 13 (Kcnj13) with primers mus4987F25 (5′-GGATGAGAGAGAGAAGCACAAGTGG-3′) and mus5140R25 (5′-CTGTATGACCAACCTTGGACATGAT-3′) served as a nuclear reference gene. All primer pairs were PCR optimized and checked by PAGE. The qPCRs for the mtDNA and *Kcnj13* were performed using the following parameters: 95 °C for 7 min, 45 cycles of 95 °C for 15 s, and 62.6 °C (*Kcnj13*) or 64.6 °C (mtDNA) for 1 min, 95 °C for 1 min, and 55 °C for 1 min.

From the obtained qPCR fluorescence data, the C_t_ values for the calculation of the mtDNA copy numbers were obtained by Chapman sigmoidal non-linear regression curve fitting analysis in Sigma Plot (2001 for Windows version 7.0, Systat Software GmbH) [100]. The shape of the sigmoidal regression curves was determined by the parameters y_0_, a, b, and c, from the equation y = y_0_ + a(1 − e^−bx^)^c^; and the inflection point of the sigmoidal curve determining the C_t_ value was calculated using the equation C_t_ = ln(c)/b. The copy number (CN) of the mtDNA relative to the diploid single nuclear *Kcnj13* was calculated as CN = 2 × 2^ΔCt^, where ΔC_t_ represents the cycle number difference between *Kcnj13* and the mtDNA fragment (ΔC_t_ = C_t_nuclear − C_t_mtDNA) [101]. The PCR amplification efficiency, (10^−1/slope^ − 1) × 100 [102], was determined as 103% and 95% for the mtDNA primer pair and the *Kcnj13* primer pair, respectively.

### 4.13. Statistical Analysis

Data analysis and statistical evaluation were performed using Excel 2016 (Microsoft) with the Excel add-in ”Real Statistics Resource Pack” (release 7.9) by Charles Zaiontz available at http://www.real-statistics.com (accessed on 5 October 2022) as well as GraphPad Prism (version 9.3.1, GraphPad Software). Since the samples were non-normally distributed in some data sets as determined by the Shapiro–Wilk normality test, and because of the limited sample sizes (number of animals), statistical significances were calculated using the non-parametric Mann–Whitney U (Wilcoxon rank-sum) test. The number of experiments, technical replicates, and significance levels for each analysis are indicated in the Figure legends.

### 4.14. Proteomic Analysis: Sample Preparation, Mass Spectrometry, and Data Analysis

Left ventricular cardiac tissue was lysed as described [103]; the muscle tissue was pulverized in liquid nitrogen, homogenized on ice in urea buffer (7 M urea, 2 M thiourea, 20 mM Tris base, pH 8.5), followed by sonication (6× for 30 s, with 30 s rest time on ice) to support protein solubilization. Protein concentration was determined by Bradford assay. Subsequent digestion was carried out as described [103]; 40 µg of proteins were digested with trypsin (ratio 1:40) in 50 mM AMBIC. Prior to digestion, reduction of cysteine bridges using 15 mM DTT was carried out at 56 °C for 30 min followed by a 30 min alkylation step using 5 mM IAA at RT. 1 µg trypsin was added to each sample and digestion was carried out overnight at 37 °C and stopped by acidification. Peptide concentration was determined by amino acid analysis [104], and 200 ng of peptides (in 0.1% TFA) were used for mass spectrometric analysis.

Mass spectrometry was carried out as described [105]; a nanoHPLC analysis was performed on an UltiMate 3000 RSLC nano LC system (Thermo Fisher Scientific, Germany). Peptides were loaded on a capillary pre-column (Thermo Fisher Scientific, 100 μm × 2 cm, particle size 5 μm, pore size 100 Å) and afterwards onto an analytical C18 column (Thermo Fisher Scientific, 75 μm × 50 cm, particle size 2 μm, pore size 100 Å). Peptide separation was performed with a linear gradient of up to 40% buffer B (84% acetonitrile, 0.1% formic acid) with a flow rate of 400 nL/min. The HPLC system was online-coupled to the nano ESI source of an Orbitrap Elite mass spectrometer (Thermo Fisher Scientific). The MS1 scan range was set from 300 to 2000 m/z and a resolution of 30,000. From each full scan, the Top 20 ions were selected for low-energy collision-induced dissociation (CID) fragmentation with a NCE of 35%. The Top 20 ions were subsequently dynamically excluded for fragmentation for 30 s.

Data analysis was carried out using MaxQuant (v.1.6.17.0) [106]. Spectra were searched against the Uniprot *Mus musculus* reference proteome (04_2021) [107] using the integrated Andromeda algorithm using trypsin as protease. The false discovery rate (FDR) was set to 1% for peptides (minimum length of 7 amino acids) and proteins and was determined by searching against a reverse decoy database. A maximum of two missed cleavages were allowed in the database search. Peptide identification was performed with an allowed initial precursor mass deviation up to 7 ppm and an allowed fragment mass deviation 20 ppm. Carbamidomethylation of cysteines was set as fixed modification and oxidation of methionine and deamidation of asparagine as variable modifications, due to sample pre-processing. For superior identification the match between run option was enabled. Quantitation was carried out using the MaxQuant Label-Free Quantification (LFQ) algorithm including unique and razor peptides for quantification. For further quantitation the calculation of iBAQ values was enabled [108]. Resulting data was subsequently statistically analyzed using Perseus (v. 1.6.14.0) [109]. Contaminants and decoys were filtered and LFQ values were log_2_-transformed. Only proteins identified in at least 4 replicates in one group were used for further statistical assessment. Remaining missing values were imputed using a width of 0.3 and a downshift of 1.8. Two-sided Student’s t-test with Benjamini–Hochberg correction was performed to determine significantly enriched proteins. Proteins with a *p*-value <0.05 or respectively adjusted *p*-value <0.05 were assigned as being significant differential between the wild-type and homozygous desmin knock-out conditions. A volcano plot that shows plots the log_10_-transformed *p*-values against the log_2_-transformed fold changes was generated using R version 4.1.0 (R Core Team, 2021) and the package ggplot2.

Hierarchical Clustering was performed to determine clusters of proteins with differential expression profiles in the wild-type and homozygous desmin knock-out conditions. For this purpose, all proteins remaining after our stringent filter criteria, were Z-scored and resulting values were averaged for both groups. For Hierarchial Clustering, Euclidean distance was chosen with average linkage and constraint and a maximum number of 300 clusters. Resulting clusters were exported and used for subsequent GO term and Pathway enrichment analysis using David Bioinformatics Resources 6.8 [110,111].

## Figures and Tables

**Figure 1 ijms-23-12020-f001:**
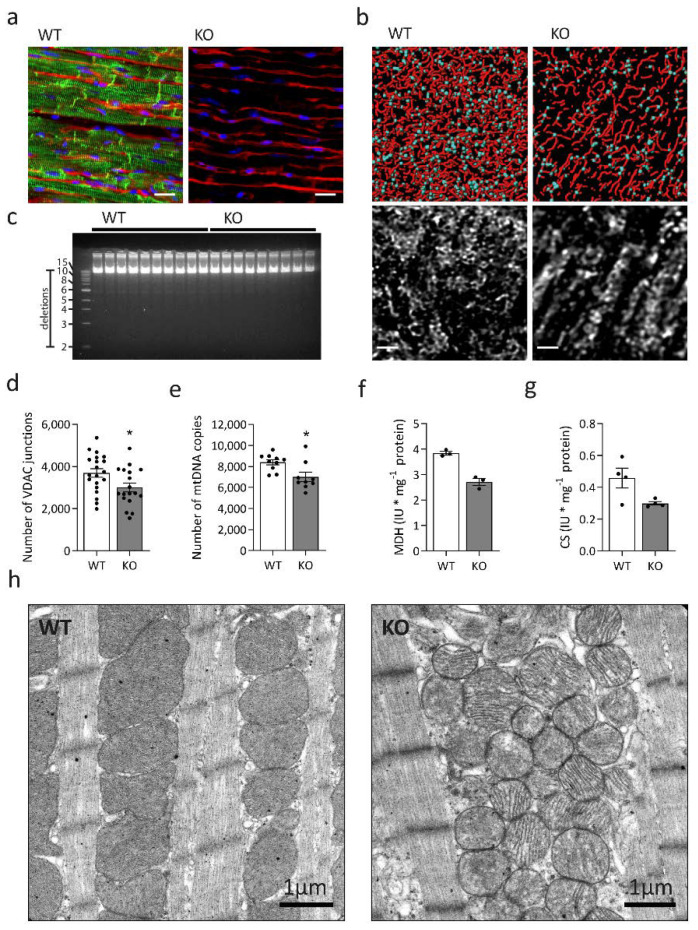
Distribution, number, and ultrastructural morphology of mitochondria in desmin knock-out cardiac tissue. (**a**) Indirect immunofluorescence images of desmin (green) in longitudinal sections of wild-type (WT) and desmin knock-out (KO) ventricular cardiac tissue in conjunction with wheat germ agglutinin conjugate (WGA-AlexaFluor647) as a sarcolemmal marker (red) and nuclear marker DAPI (blue). Scale bars, 20 µm. (**b**) Analysis of the three-dimensional distribution of mitochondria in left ventricular cardiac tissue. Sections derived from six-month-old desmin knock-out mice (KO) and wild-type littermates (WT) were stained with an antibody directed against the voltage-dependent anion channel (VDAC1) (lower panel, greyscale). Confocal Z-stack images from longitudinally oriented cardiomyocytes were processed to visualize the mitochondrial network (upper panel, red). As a surrogate marker for mitochondrial contact sites, pixels that label crossovers from different mitochondrial fluorescence signals were determined (upper panel, light blue dots). Scale bars, 2 µm. (**c**) Long-range PCRs from total DNA extracted from left ventricular myocardium derived from desmin knock-out mice and wild-type littermates did not show large-scale mtDNA deletions. (**d**) Quantitation of VDAC1-positive mitochondrial contact sites revealed a significantly reduced number indicating a rarefication of the mitochondrial network in desmin knock-out cardiomyocytes. Five Z-stacked images of different regions of cardiac muscle tissue specimens from each of the four animals of each genotype were analyzed; Mann-Whitney test, * *p* < 0.05. (**e**) Assessment of mitochondrial DNA (mtDNA) copy numbers by quantitative PCR confirmed a significant decrease in desmin knock-out mice. Samples from nine homozygous and ten wild-type mice were analyzed in nine technical replicates: Mann-Whitney test, * *p* < 0.05. (**f**,**g**) Spectrophotometrically determined enzyme activities of malate dehydrogenase (MDH; samples from three animals per genotype analyzed in singlet; Mann-Whitney test, not significant) and citrate synthase (CS; samples from four animals per genotype analyzed in singlet; Mann-Whitney test, not significant) in left ventricular cardiac tissue homogenates were non-significantly reduced in the knock-out condition. Values in (**d**–**g**) are given as mean ± SEM; (**h**) Electron microscopy depicted areas of focal clustering of mitochondria in conjunction with a marked coarsening of the mitochondrial cristae in the cardiac tissue of desmin knock-out mice (KO) as compared to the wild-type (WT).

**Figure 2 ijms-23-12020-f002:**
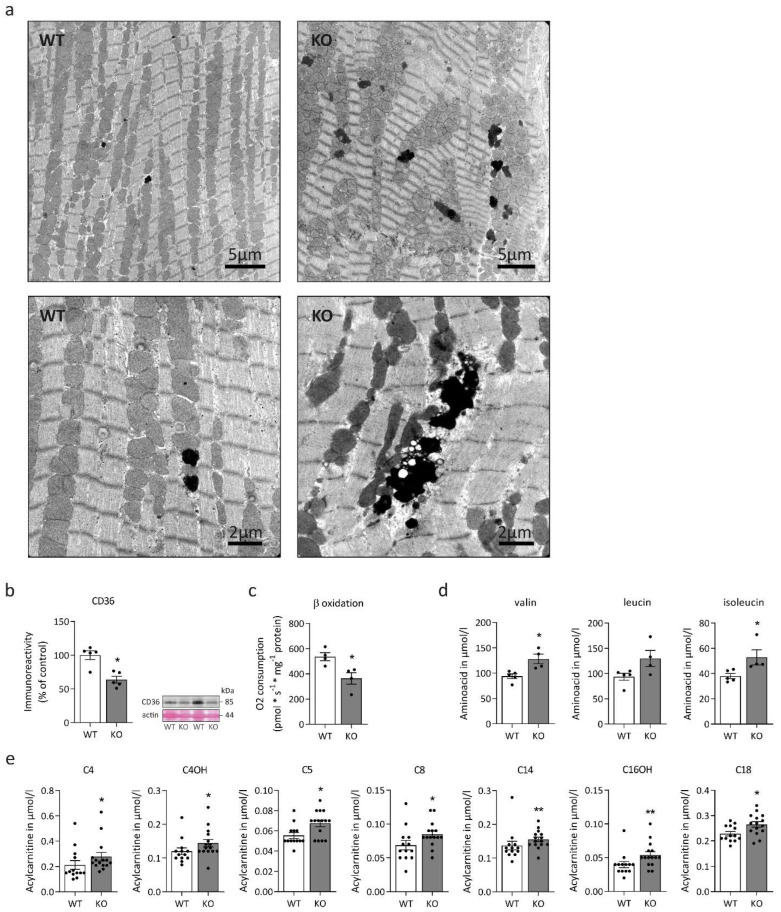
Morphological and biochemical aspects of fatty and amino acid metabolism in desmin knock-out cardiac tissue. (**a**) Compared to cardiac tissue from wild-type siblings (WT), ultrastructural analysis of desmin knock-out mice (KO) demonstrated a marked increase of intermyofibrillar, electron-dense material corresponding to lipofuscin deposits. (**b**) Immunoblot analysis of protein levels of the fatty acid transporter CD36 in left ventricular cardiac tissue homogenates depicted a significant reduction in desmin knock-out mice. Samples from five animals per genotype were analyzed in duplicate: Mann-Whitney test, * *p* < 0.05. Shown is a representative CD36 immunoblot in conjunction with the Ponceau S-stained membrane as loading control. The apparent molecular weight of CD36 is 85 kDa. (**c**) Mitochondrial respiration rate of beta-oxidation stimulated by octanoyl-carnitine was significantly lower in desmin knock-out mice (samples from four animals per genotype were analyzed in duplicate: Mann-Whitney test, * *p* < 0.05). (**d**) Mass spectrometry-based quantitation of amino acids in dried whole blood samples denoted increased levels of valine, isoleucine, and leucine; the latter just failed to reach statistical significance (Table 1). Samples from four homozygous and five wild-type mice were analyzed in singlet; Mann-Whitney test, * *p* < 0.05. (**e**) Mass spectrometry-based quantitation of acylcarnitines in dried whole blood samples revealed increased concentrations of multiple acyl-carnitines, i.e., butyryl- and hydroxy-butyryl-carnitine (C4, C4OH), isovaleryl-carnitine (C5), octanoyl-carnitine (C8), tetradecanoyl-carnitine (C14), hydroxy-hexadecanoyl-carnitine (C16OH), and octanoyl-carnitine (C18) in desmin knock-out mice. In addition, levels of propionyl-carnitine (C3), palmitoyl-carnitine (C16), 3OH-hexadecenoyl-carnitine (C16:1OH), and oleoyl-carnitine (C18:1) were also elevated, however, they just failed to reach statistical significance (Table 1). Samples from fifteen homozygous and thirteen wild-type mice were analyzed in singlet; Mann-Whitney test, * *p* < 0.05, ** *p* < 0.01. Values in (**b**–**e**) are given as mean ± SEM.

**Figure 3 ijms-23-12020-f003:**
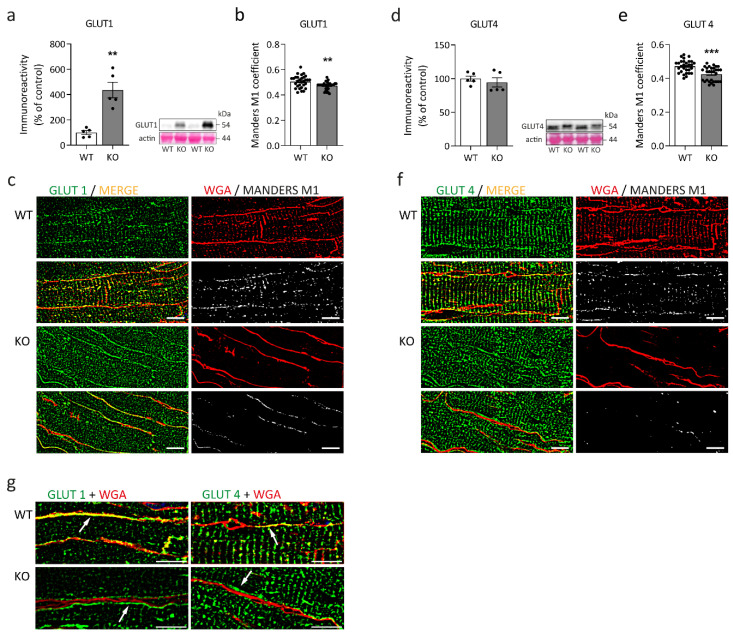
Expression and localization of glucose transporters in desmin knock-out cardiac tissue. (**a**,**d**) Immunoblot analysis addressing the expression of GLUT1 (**a**) and GLUT4 (**d**) in left ventricular cardiac tissue homogenates. Samples from five animals per genotype were immunoblotted in duplicate; Mann-Whitney test, ** *p* < 0.01. Representative immunoblots and Ponceau S-stained membranes are shown. The apparent molecular weight of GLUT1 and GLUT4 is 54 kDa. Note the highly increased expression of GLUT1 as well as the slight shift of the GLUT4 immunoblot signals to a higher molecular weight in desmin knock-out hearts. (**b**,**e**) Quantitative analysis of the colocalization of both glucose transporters with the sarcolemma based on the Manders M1 coefficient resulted in a statistically significant difference between desmin knock-out and wild-type animals (ten images of different regions of cardiac muscle tissue specimens from each of the three animals of each genotype were analyzed; Mann-Whitney test, ** *p* < 0.01, *** *p* < 0.001). Values in (**a**,**b**,**d**,**e**) are given as mean ± SEM. (**c**,**f**) The subcellular localization of GLUT1 ((**c**), green) and GLUT4 ((**f**), green) in relation to the sarcolemma stained by Wheat Germ Agglutinin (WGA, red) was visualized in longitudinally oriented cardiomyocytes. Note the absence of the WGA signal within the t-tubules region of sarcoplasm of the desmin knock-out cardiomyocytes (upper right images of KO). Black and white images represent colocalized pixels according to Manders M1 coefficient (white). (**g**) Qualitative examination depicted a localization of both GLUT1 and GLUT4 underneath the sarcolemma in desmin knock-out cardiomyocytes (white arrows, lower images), whereas the signals of both glucose transporters in wild-type tissue colocalized to the WGA-stained sarcolemma (white arrows, upper images). Scale bars, 10 μm.

**Figure 4 ijms-23-12020-f004:**
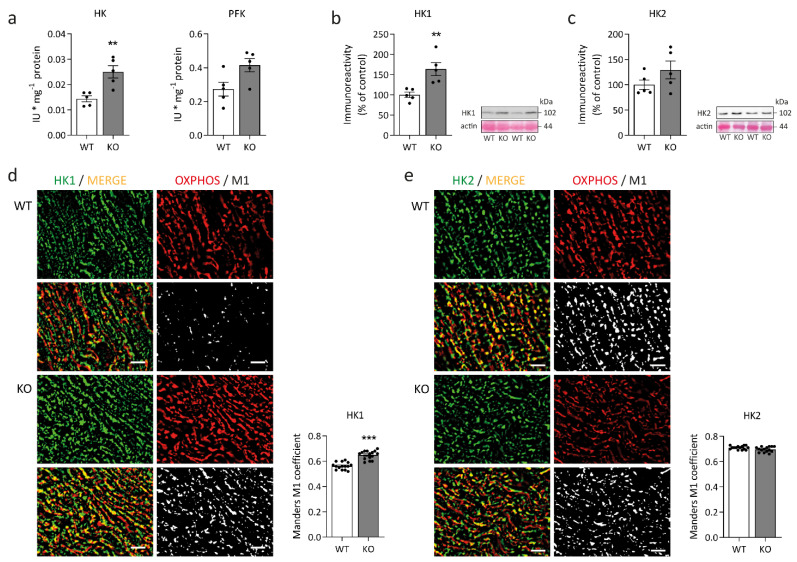
Activities, protein levels, and localization of glucose metabolism rate-limiting enzymes in desmin knock-out cardiac tissue. (**a**) Spectrophotometrically-determined enzyme activities of hexokinase (HK), which was significantly elevated, and phosphofructokinase (PFK) in desmin knock-out mice (KO) and their wild-type siblings (WT). For both analyses, left ventricular cardiac tissue homogenates from five animals per genotype were analyzed in duplicate; Mann-Whitney test, ** *p* < 0.01). (**b**,**c**) Further immunoblot analysis addressing hexokinase isoforms 1 (HK1) and 2 (HK2) protein levels in the homogenates revealed a statistically significant increase in the HK1 amount. For both HK1 and HK2 homogenates from five animals per genotype were analyzed in quadruplicate (HK1) or triplicate (HK2); Mann-Whitney test, ** *p* < 0.01. Representative immunoblots and Ponceau S-stained membranes are shown. The apparent molecular weight of HK1 and HK2 is 102 kDa. (**d**,**e**) The subcellular localization of HK1 and HK2 (green) in relation to mitochondrial OXPHOS components (red) was visualized in longitudinally oriented cardiomyocytes. The additional black and white images illustrate pixels of colocalization (white). Scale bars, 5 μm. Quantitative analysis based on the Manders M1 coefficient determined an increased degree of colocalization of HK1 with the mitochondrial compartment in desmin knock-out mice (five images of different regions of cardiac muscle tissue specimens from each of the three animals of each genotype were analyzed; Mann-Whitney test, *** *p* < 0.001), while analysis of HK2 did not show any difference. Values in (**a**–**e**) are given as mean ± SEM.

**Figure 5 ijms-23-12020-f005:**
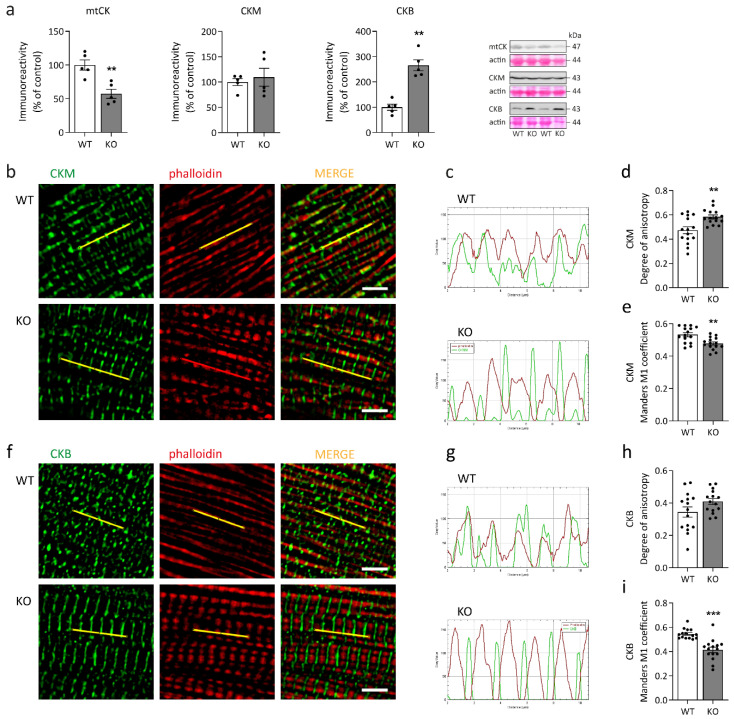
Creatine kinase protein levels and localization in desmin knock-out cardiac tissue. (**a**) Immunoblot analysis addressing the protein levels of mitochondrial creatine kinase (mtCK) as well as the cytosolic CKM and CKB in left ventricular cardiac tissue homogenates from desmin knock-out mice (KO) and their wild-type siblings (WT). Note the significantly lower protein level of mtCK and the higher level of CKB in desmin knock-out mice. Representative immunoblots and Ponceau S-stained membranes are shown. The apparent molecular weight of both CKM and CKB is 43 kDa and of mtCK 47 kDa. For each CK isoform, homogenates from five animals per genotype were analyzed in duplicate; Mann-Whitney test, ** *p* < 0.01. (**b**,**f**) The subcellular localization of CKM and CKB (green) in relation to actin filaments (red) was visualized in longitudinally oriented cardiomyocytes. Scale bar, 5 μm. (**c**,**g**) Fluorescence intensity line profiles of both creatine kinases (lines indicated in yellow in panels (**b**) and (**f**)) demonstrated a phase-shift along the longitudinal axis of the myofibrils in the desmin knock-out cardiac tissue. (**d**,**h**) The degree of anisotropy of CKM (**d**) and CKB (**h**) fluorescence signals, which addresses the regularity of the striated signal patterns, showed only a significant increase in the case of CKM (five images of different regions of cardiac muscle tissue specimens from each of the three animals of each genotype were analyzed; Mann–Whitney test, ** *p* < 0.01). (**e**,**i**) Quantitative analysis based on the Manders M1 coefficient determined a significantly decreased colocalization of both CKM (e) and CKB (i) with actin filaments in the desmin knock-out genotype (five images of different regions of cardiac muscle specimens from each of the three animals of each genotype were analyzed; Mann-Whitney test, ** *p* < 0.01, *** *p* < 0.001). Values in (**a**,**d**,**e**,**h**,**i**) are given as mean ± SEM.

**Figure 6 ijms-23-12020-f006:**
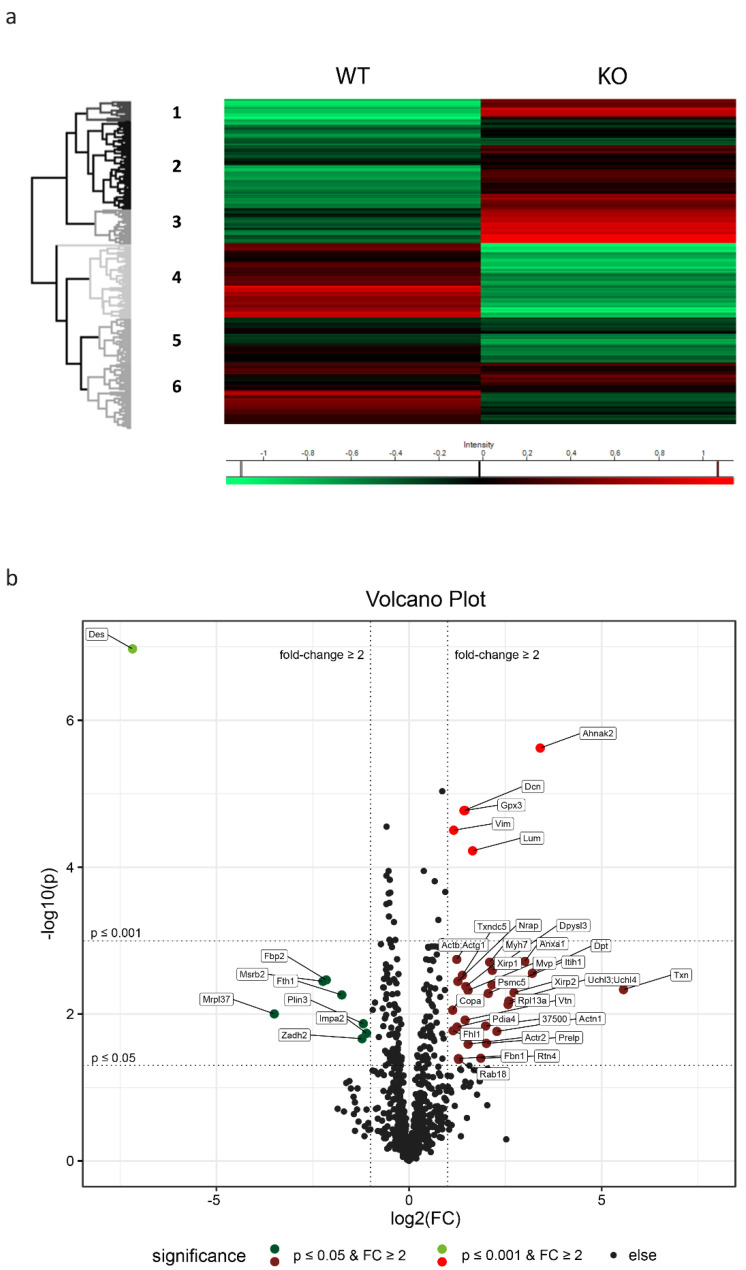
Quantitative proteomic analysis of left ventricular cardiac tissue derived from desmin knock-out mice. (**a**) Hierarchical clustering of the proteomic data created a dendrogram with six distinct clusters. Clusters 1 to 3 showed higher protein amounts in the desmin knock-out cardiac tissue, while clusters 4 to 6 highlighted reduced protein amounts as compared to their wild-type sibling derived tissue. The heat map indicates increased protein levels in red and decreased values in green. For a list of all proteins used for hierarchical clustering please refer to Appendix A; for a bubble plot illustration of the “Gene Ontology (GO) term Cellular compartment (CC)” and “Kyoto Encyclopedia of Genes and Genomes (KEGG) pathway” enrichments please refer to Appendix A. (**b**) Volcano plot comparing protein levels of desmin knock-out and wild-type genotypes. *X*-axis, log_2_-transformed mean fold change; *y*-axis, -log_10_-transformed *p*-value. Light red and dark red as well as light and dark green dots indicate significantly up- and down-regulated proteins, respectively, with *p*-values of ≤ 0.05 or ≤ 0.001 and a fold-change ≥ 2. For lists of significantly up- and downregulated proteins as well as all quantified proteins including non-significantly regulated and proteins of any fold-change in desmin knock-out left ventricular cardiac tissue please refer to Appendix A.

**Table 1 ijms-23-12020-t001:** Acylcarnitine concentrations were determined from dried blood sample cards derived from fifteen homozygous desmin knock-out mice and thirteen wild-type littermates; in addition, acylcarnitine ratios were calculated. Amino acid concentrations were derived from four homozygous and five wild-type mice. In a few samples, concentrations of specific acylcarnitines were below the detection limit resulting in lower sample numbers (n, as indicated).

Acylcarnitines in µmol/L	Desmin Knock-out (Mean ± SEM; *n*)	Wild-Type Littermate (Mean ± SEM; *n*)	*p-*Value (Two-Tailed) (Mann–Whitney Test)
Carnitine (C0)	17.52 ± 0.54; 15	16.79 ± 0.63; 13	0.434
Acetyl-carnitine (C2)	15.09 ± 0.84; 15	13.02 ± 0.38; 13	0.140
**Propionyl-carnitine (C3)**	0.51 ± 0.02; 15	0.45 ± 0.02; 13	**0.059**
Malonyl-carnitine (C3DC)	0.14 ± 0.01; 14	0.14 ± 0.02; 11	0.298
**Butyryl-carnitine (C4)**	0.28 ± 0.03; 15	0.21 ± 0.04; 13	**0.036**
Methylmalonyl-carnitine (C4DC)	undetectable	undetectable	―
**3OH-Butyryl-carnitine (C4OH)**	0.14 ± 0.01; 15	0.12 ± 0.01; 13	**0.027**
**Isovaleryl-carnitine (C5)**	0.07 ± 0.00; 15	0.06 ± 0.00; 13	**0.023**
Tiglyl-carnitine (C5:1)	undetectable	undetectable	―
Glutaryl-carnitine (C5DC)	0.07 ±0.01; 15	0.07 ± 0.01; 11	0.938
3OH-Isovaleryl-carnitine (C5OH)	undetectable	undetectable	―
Hexanoyl-carnitine (C6)	0.07 ± 0.00; 15	0.07 ± 0.01; 13	0.645
**Octanoyl-carnitine (C8)**	0.08 ± 0.00; 15	0.07 ± 0.01; 13	**0.030**
Octenoyl-carnitine (C8:1)	undetectable	undetectable	―
Decanoyl-carnitine (C10)	0.04 ± 0.00; 15	0.04 ± 0.00; 13	0.447
Cis4-Decanoyl-carnitine (C10:1)	0.03 ± 0.00; 10	0.02 ± 0.00; 11	0.139
Dodecanoyl-carnitine (C12)	0.21 ± 0.01; 15	0.19 ± 0.02; 13	0.214
**Tetradecanoyl-carnitine (C14)**	0.16 ± 0.01; 15	0.14 ± 0.01; 13	**0.010**
Tetradecenoyl-carnitine (C14:1)	0.07 ± 0.00; 15	0.07 ± 0.00; 13	0.369
Tetradecadienyl-carnitine (C14:2)	0.03 ± 0.00; 15	0.02 ± 0.00; 13	0.475
3OH-Tetradecanoyl-carnitine (C14OH)	0.02 ± 0.00; 15	0.01 ± 0.00; 13	0.174
**Palmitoyl-carnitine (C16)**	1.00 ± 0.04; 15	0.92 ± 0.04; 13	**0.088**
Palmitoleyl-carnitine (C16:1)	0.08 ± 0.00; 15	0.08 ± 0.01; 13	0.279
**3OH-Palmitoyl-carnitine (C16OH)**	0.05 ± 0.00; 15	0.04 ± 0.00; 13	**0.012**
**3OH-Hexadecenoyl-carnitine (C16:1OH)**	0.032 ± 0.002; 15	0.026 ± 0.003; 13	**0.076**
**Octadecanoyl-carnitine (C18)**	0.27 ± 0.01; 15	0.23 ± 0.01; 13	**0.025**
**Oleoyl-carnitine (C18:1)**	0.34 ± 0.02; 15	0.30 ± 0.02; 13	**0.065**
3OH-Stearoyl-carnitine (C18OH)	0.01 ± 0.00; 14	0.01 ± 0.00; 11	0.584
3OH-Oleoyl-carnitine (C18:1OH)	0.02 ± 0.00; 15	0.02 ± 0.00; 13	0.645
3OH-Linolyl-carnitine (C18:2OH)	0.03 ± 0.00; 15	0.03 ± 0.00; 13	0.333
**Acylcarnitine ratios**	**Desmin knock-out** ** (mean ± SEM; n)**	**Wild-type littermate** **(mean ± SEM; n)**	***p-*Value (one-tailed)** **(Mann–Whitney test)**
C3/C0	0.03 ± 0.00; 15	0.03 ± 0.00; 13	0.099
C3/C2	0.03 ± 0.00; 15	0.03 ± 0.00; 13	0.418
C5/C2	0.00 ± 0.00; 15	0.00 ± 0.00; 13	0.238
C8/C2	0.01 ± 0.00; 15	0.01 ± 0.00; 13	0.111
**C8/C10**	2.12 ± 0.11; 15	1.85 ± 0.12; 13	**0.049**
C8/C12	0.41 ± 0.02; 15	0.40 ± 0.06; 13	0.191
C14:1/C14	0.48 ± 0.02; 15	0.52 ± 0.04; 13	0.282
C14:1/C16	0.08 ± 0.00; 15	0.07 ± 0.00; 13	0.356
**Amino acids in µmol/l**	**Desmin knock-out** **(mean ± SEM; n)**	**Wild-type littermate** **(mean ± SEM; n)**	***p-*Value (two-tailed)** **(Mann–Whitney test)**
Alanine	303 ± 51; 4	256 ± 31; 5	0.806
Allo-Isoleucine	<2.0; 4	<2.0; 5	―
Arginine	25.1 ± 2.4; 4	25.4 ± 2.3; 5	0.624
Citrulline	13.6 ± 1.3; 4	14.9 ± 1.8; 5	1.000
Glutamine	107.2 ± 16.7; 4	94.6 ± 18.0; 5	0.462
Glycine	64.9 ± 6.0; 4	53.3 ± 5.5; 5	0.221
Histidine	3.54 ± 0.83; 4	2.51 ± 0.30; 5	0.327
**Isoleucine**	52.9 ± 6.0; 4	38.0 ± 2.0; 5	**0.014**
**Leucine**	129.9 ± 16.1; 4	93.7 ± 6.9; 5	**0.086**
Lysine	20.1 ± 1.8; 4	19.1 ± 2.2; 5	1.000
Methionine	12.9 ± 1.0; 4	11.8 ± 1.4; 5	1.000
3-O-Methyldopa	0.063 ± 0.003; 4	0.056 ± 0.002; 5	0.178
Ornithine	2.6 ± 0.6; 4	<2.0; 5	―
**Phenylalanine**	59.0 ± 12.2; 4	44.6 ± 8.1; 5	**0.086**
Proline	93.3 ± 16.0; 4	73.9 ± 12.1; 5	0.327
Serine	83.5 ± 10.4; 4	62.6 ± 9.6; 5	0.221
Threonine	52.9 ± 1.5; 4	44.7 ± 4.9; 5	0.221
**Tryptophan**	3.47 ± 0.44; 4	2.45 ± 0.23	**0.086**
Tyrosine	59.1 ± 12.7; 4	50.6 ± 9.5; 5	0.327
**Valine**	128.0 ± 9.6; 4	94.1 ± 4.7; 5	**0.014**

## Data Availability

Raw data and a method description have been deposited to the ProteomeXchange Consortium via the PRIDE partner repository [41] (https://www.ebi.ac.uk/pride) with the dataset identifier PXD030938.

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
