# Peer review of "Desmin Knock-Out Cardiomyopathy: A Heart on the Verge of Metabolic Crisis"

_ijms, 2022, doi:10.3390/ijms231912020_

Round 1

Reviewer 1 Report

The manuscript presented by Elsinocova B. et al concerns the cardiac metabolic aspect of desmin knock-out mice. The research design is appropriate and work is performed correctly. The data bring with some new aspects in desmin related myopathy and will be interesting for the researchers in this field.  However, some questions will be addressed before the publication.

1-      The localization of Glut1 and Glut4 is altered compared to control cardiac cells, but no further discuss with this observation.  Authors should discuss the potential effect on glucose transport, and if it is compatible with the increased glucose metabolism as supposed by authors, given the transport of glucose could be affected. Similar situation for creatine kinase B and MT-CK, which change possible for energy metabolism?

2-      Authors used the more recent mass spectrometric analytical method to study the proteomic profile, given the restrictions of each method for proteins analysis, it is acceptable the data will be not fit well between the methods. It is not sure which is really right without furthermore other characterization. It is more appropriate to remove the sentence “ as time passed on technically more advanced analysis”.

3-      Concerning the study of acylcarnitine study from dried blood samples, it should be indicated the limit of this data at least in discussion in the case of absence of data from skeletal muscles, given the more important mass of skeletal muscle compared to heart

Author Response

Reviewer 1

The manuscript presented by Elsnicova B. et al concerns the cardiac metabolic aspect of desmin knock-out mice. The research design is appropriate and work is performed correctly. The data bring with some new aspects in desmin related myopathy and will be interesting for the researchers in this field. However, some questions will be addressed before the publication.

Comment 1:

The localization of Glut1 and Glut4 is altered compared to control cardiac cells, but no further discuss with this observation. Authors should discuss the potential effect on glucose transport, and if it is compatible with the increased glucose metabolism as supposed by authors, given the transport of glucose could be affected. Similar situation for creatine kinase B and MT-CK, which change possible for energy metabolism?

We followed this valuable suggestion and accordingly added information to the Discussion paragraph “4.4 Metabolic adaptations in desmin knock-out hearts”; for details, please see the tracked changes within the revised manuscript. In our opinion, the observed GLUT1 changes are compatible with increased compensatory glucose metabolism. Regarding creatine kinases, in particular, the increased amounts of the more ROS-resistant CKB possessing a higher affinity to phosphocreatine in conjunction with the shifts of CKM and CKB towards the levels of sarcomeric M-lines suggest enhanced metabolic channeling of ATP to myosin-ATPase. The reduction of mtCK, and thus mitochondria-related phosphocreatine formation, can be an important limiting step in phosphocreatine turnover in the desmin knock-out heart under the increased load. This harm can be compensated by the enhancement of glycolysis, which we observe in desmin knock-out hearts.

Comment 2:

Authors used the more recent mass spectrometric analytical method to study the proteomic profile, given the restrictions of each method for proteins analysis, it is acceptable the data will be not fit well between the methods. It is not sure which is really right without furthermore other characterization. It is more appropriate to remove the sentence “as time passed on technically more advanced analysis”.

We removed the entire sentence, “All these findings at least partially do not fit with our – as time passed on – technically more advanced analysis.”, from the Discussion section.

Comment 3:

Concerning the study of acylcarnitine study from dried blood samples, it should be indicated the limit of this data at least in discussion in the case of absence of data from skeletal muscles, given the more important mass of skeletal muscle compared to heart.

At the first mention of the acylcarnitine findings in the Discussion section, “Notably, our analysis of dried whole blood samples depicted an intriguing picture in which multiple acylcarnitines ranging from C3 to C18 chain length showed significantly increased concentrations in desmin knock-out mice.”, we already had added the information and ‘attenuation’ of this results by the subsequent sentence, “Regarding this finding, one has to keep in mind that the levels of blood acylcarnitines are determined by a far greater extent by the skeletal than the cardiac muscle metabolism.”, to which we now added the word “far”. We hope that settles the matter.

Reviewer 2 Report

The authors have conducted a very interesting study, with a good design  that integrates two different approaches to the mechanism of the disease. The results are clearly presented, and it is very easy to follow their presentation, I would like to specially congratulate the authors on their exquisite and representative images. Another aspect that I quite enjoyed of this work is the section of the translational aspects that are derived from this study. Congratulations 

Author Response

We thank the referee for this extraordinarily positive evaluation of our work.

Reviewer 3 Report

The authors evaluated the left ventricular cardiac muscle in adult desmin deficient mice using a variety of approaches (including morphological, clinical chemistry, biochemical, genetic, and proteomic methods), and found that loss of desmin in mouse cardiac muscle results in remarkable myocardial metabolic dysfunction, including altered glucose, fatty acid, and amino acid metabolism.

Elsnicova and colleagues have written an interesting and scientifically significant study in which they further assess the cardiac outcome in adult desmin deficient mice. These findings agree with previous works and provide additional important information regarding desminopathy that was not previously reported. Overall, the manuscript is novel and well-written, but the findings are mostly descriptive or correlational and would benefit from improved rigor. I have some minor comments.

1)    Bar charts should be used with all individual data points visualized throughout.

2)    2.2 patients in Materials and methods: please add the Institutional Review Board number if available.

3)    Font format is not consistent. For example, Page 10 line 42, 43 and 46-48. Please correct it.

4)    Result part: 3.4 Proteome analysis: Page 18 line 17, I don’t think you need to put your account name and password in the manuscript.

5)    The description of Proteome analysis should be organized. The authors could use different subtitles to present it in a clear way. Some of the descriptions can be put in the method part. Also, is there any validation of the proteomic findings?  

Author Response

The authors evaluated the left ventricular cardiac muscle in adult desmin deficient mice using a variety of approaches (including morphological, clinical chemistry, biochemical, genetic, and proteomic methods), and found that loss of desmin in mouse cardiac muscle results in remarkable myocardial metabolic dysfunction, including altered glucose, fatty acid, and amino acid metabolism.

Elsnicova and colleagues have written an interesting and scientifically significant study in which they further assess the cardiac outcome in adult desmin deficient mice. These findings agree with previous works and provide additional important information regarding desminopathy that was not previously reported. Overall, the manuscript is novel and well-written, but the findings are mostly descriptive or correlational and would benefit from improved rigor. I have some minor comments.

Comment 1:

Bar charts should be used with all individual data points visualized throughout.

We followed this suggestion and updated all graphs within all figures accordingly.

Comment 2:

2.2 patients in Materials and methods: please add the Institutional Review Board number if available.

This number, #20922, has been added to the Text.

Comment 3:

Font format is not consistent. For example, Page 10 line 42, 43 and 46-48. Please correct it.

Done. This was not the case in our originally submitted Word document, but the result of the Journal’s editing process. We checked the entire actual document for format issues and performed corrections.

Comment 4:

Result part: 3.4 Proteome analysis: Page 18 line 17, I don’t think you need to put your account name and password in the manuscript.

We removed the account information from the manuscript, which was only provided for the purpose of the reviewing process. After acceptance for publication of the manuscript, the PRIDE repository dataset PXD030938 will be made public.

Comment 5:

The description of Proteome analysis should be organized. The authors could use different subtitles to present it in a clear way. Some of the descriptions can be put in the method part. Also, is there any validation of the proteomic findings?

For better readability and improvement of the public perception of the data presentation, we added, as suggested, subsequently to the original subtitle, “Proteome analysis of left ventricular cardiac tissue reveals widespread alterations related to subcellular compartments and metabolism”, two more subtitles, namely, “The pattern of significantly regulated proteins in desmin knock-out cardiac tissue” and “A closer look at metabolism-related proteins in desmin knock-out cardiac tissue”, to the Results section.

Additional comment:

Moderate English changes required.

We improved English language and style throughout the entire manuscript text.